# Implications of the differing roles of the β1 and β3 transmembrane and cytoplasmic domains for integrin function

Zhenwei Lu[1†], Sijo Mathew[2†], Jiang Chen[1†‡], Arina Hadziselimovic[1], Riya Palamuttam[1], Billy G Hudson[1,2,3,4], Reinhard Fässler[5], Ambra Pozzi[2,6,7,8], Charles R Sanders[1*], Roy Zent[2,3,6,7*]

[1]Department of Biochemistry, Vanderbilt University Medical Center, Nashville, United States; [2]Division of Nephrology, Department of Medicine, Vanderbilt Medical Center, Nashville, United States; [3]Department of Cell and Developmental Biology, Vanderbilt University Medical Center, Nashville, United States; [4]Department of Pathology, Microbiology and Immunology, Vanderbilt University Medical Center, Nashville, United States; [5]Department of Molecular Medicine, Max Planck Institute of Biochemistry, Martinsried, Germany; [6]Department of Cancer Biology, Vanderbilt University Medical Center, Nashville, United States; [7]Veterans Affairs Hospital, Nashville, United States; [8]Department of Molecular Physiology and Biophysics, Vanderbilt University Medical Center, Nashville, United States

**\*For correspondence:** chuck.
sanders@vanderbilt.edu (CRS);
roy.zent@vanderbilt.edu (RZ)

†These authors contributed
equally to this work

**Present address:** ‡Department
of Biomedical Sciences and
Pathobiology, Virginia Tech,
Blacksburg, United States

**Competing interest:** See
page 27

**Reviewing editor:** Maddy
Parsons, King's College London,
United Kingdom

**Abstract** Integrins are transmembrane receptors composed of α and β subunits. Although most integrins contain β1, canonical activation mechanisms are based on studies of the platelet integrin, αIIbβ3. Its inactive conformation is characterized by the association of the αIIb transmembrane and cytosolic domain (TM/CT) with a tilted β3 TM/CT that leads to activation when disrupted. We show significant structural differences between β1 and β3 TM/CT in bicelles. Moreover, the 'snorkeling' lysine at the TM/CT interface of β subunits, previously proposed to regulate αIIbβ3 activation by ion pairing with nearby lipids, plays opposite roles in β1 and β3 integrin function and in neither case is responsible for TM tilt. A range of affinities from almost no interaction to the relatively high avidity that characterizes αIIbβ3 is seen between various α subunits and β1 TM/CTs. The αIIbβ3-based canonical model for the roles of the TM/CT in integrin activation and function clearly does not extend to all mammalian integrins.

## Introduction

Integrins are the principal receptors of cells for the extracellular matrix (ECM) and are heterodimeric transmembrane proteins consisting of α and β subunits (*Hynes, 2002*; *Mathew et al., 2012a*; *Pozzi and Zent, 2013*). The 18 α and 8 β integrin subunits associate non-covalently to form 24 heterodimeric integrins, which are expressed in different combinations on all cell types throughout the body. All integrin subunits, except β4, have a large extracellular ligand-binding domain, a single transmembrane domain (TM) that spans the cell membrane and a short cytosolic C-terminal domain (CT) that regulates integrin function by binding with cytoskeletal and signaling proteins. β1 integrins, in which the β1 subunit can be paired with any one of 12 different α isoforms, are the principal integrins found in solid organs.

For integrins to mediate cell adhesion to the ECM and transduce signals from the outside to the inside of the cell and vice-versa they must be able to respond to ligand binding and must also be

**eLife digest** Proteins called integrins span the membranes of most human cells, and help our cells to interact with their surroundings, enabling them to organise, communicate and to form a variety of structures. Cells in different parts of the body typically produce different integrins so that they can specifically connect with other cells and proteins in their local environment. There are many different kinds of integrin proteins found in cell membranes and they consist of one alpha and one beta subunit.

Different integrin pairs can have different effects based on their environment and the other molecules that they encounter. Much of the research into how integrins work has involved one specific integrin found in platelets – cells in blood that aid clotting and wound repair. Yet, it is unknown if all integrins actually operate in the same way as the platelet integrin.

Lu, Mathew, Chen et al. studied the part of integrins that are located inside cells (referred to as the cytoplasmic tail) and the part that crosses the membrane (the transmembrane domain). Three-dimensional structures of these parts of the proteins showed that they varied between different beta integrin proteins. Further experiments revealed that the strength of the association between different alpha and beta integrins also varied. Finally Lu, Mathew, Chen et al. demonstrated that components shared by several beta integrins actually have different purposes in different contexts.

The diversity of structures and interactions within the group of integrin proteins suggests that integrins are likely to behave very differently in different cells. This means that platelet integrins cannot be used to fully understand the activity of all other types of integrin. More work is now needed to understand how the differences between integrins affect the roles that they fulfil and the molecules that they interact with. A deeper understanding of the differences between integrins could ultimately shape the development of strategies to specifically target them to treat a range of diseases – such as cancer and diseases in which there is a build-up of fibrous connective tissue.

modulatable in terms of ligand binding affinity. The paradigm for regulation of integrin affinity is set by the platelet-specific integrin αIIbβ3 found in the inactive state under normal conditions and activated to mediate platelet adhesion to fibrinogen following injury (*Xiao et al., 2004*) (*Coller and Shattil, 2008*). In the low affinity 'inactive' state integrin αIIbβ3 has a bent conformation that is stabilized by an outer clasp involving contacts between αIIb and β3 located near the middle and ecto-plasmic ends of their TMs and an inner clasp involving αIIb/β3 contacts located in the cytosolic juxtamembrane domains, the latter of which includes a critical salt bridge. Within the inactive state heterodimer, the αIIb transmembrane helix is thought be reasonably well-aligned with the bilayer, while the long TM helix of β3 is thought to have a pronounced (25°) tilt angle that is required to maintain simultaneous formation of the outer and inner clasps (*Lau et al., 2009*; *Yang et al., 2009*; *Zhu et al., 2009*).

It has been proposed that the tilt of the β3 TM in the inactive state is promoted by a 'snorkeling' interaction of the side chain $\varepsilon$-$NH_3^+$ moiety of the membrane-buried lysine-716 with a negatively charged lipid phosphodiester group located at the adjacent membrane-water interface (*Kim et al., 2012*). Following platelet activation, integrin αIIbβ3 is converted to a high affinity state that is mediated by the binding of intracellular cytoplasmic proteins to the CT of β3. The best studied of these interactions is the binding of the cytoskeletal protein talin to the β CT. Talin is a key regulator of integrin activation and is comprised of head and rod domains. The head has an atypical FERM (band 4.1, ezrin, radixin, and moesin) domain containing four subdomains: F0-F3. F3 contains a phospho-tyrosine-binding (PTB) signature that binds to the NPxY motif shared by both integrin β1 and β3 CT, while F0-F2 enhances binding through favorable electrostatic interactions with anionic lipids in the vicinity of the integrin on the membrane surface (*Moore et al., 2012*; *Saltel et al., 2009*; *Anthis et al., 2010*). This leads to a reduction of the tilt angle of the β3 TM and destabilization of the inner membrane clasp between the β3 and αIIb subunits, promoting dissociation of the trans-membrane domains and integrin activation (*Ye et al., 2014*). Whether kindlins, a second FERM domain containing protein family with key roles in integrin activation, also contribute to the TM

dissociation is an area of active investigation (*Ye et al., 2013*; *Theodosiou et al., 2016*; *Moretti et al., 2013*; *Lefort et al., 2012*; *Montanez et al., 2008*; *Moser et al., 2009a*, *2009b*, *2008*).

Despite extensive knowledge about the structural basis of αIIbβ3 integrin activation, data is limited as to whether the widely expressed β1 containing integrins also adopt both inactive and affinity-modulated active states and, if so, how this happens. While there is evidence that the fibronectin α5β1 integrin may resemble αIIbβ3 (*Takagi et al., 2003*; *Luo et al., 2004*), it is unclear whether all the 12 possible αβ1 integrin combinations are activated similarly to αIIbβ3, or whether there is mechanistic heterogeneity underlying their various adhesive functions. That the paradigm of integrin activation established by αIIbβ3 might extend to β1 integrins is suggested by fairly high sequence homology between the β1 and β3 integrin TMs and CTs (*Figure 1*). In addition, the putative snorkeling lysine K752 of β1 integrin was shown to regulate activation of integrin α5β1 in the same manner as proposed for K716 of β3 (*Kim et al., 2012*). By contrast, disruption of the inner membrane clasp in β1 integrin did not cause any phenotype in knockin mice, nor did it alter integrin activation as assessed by activation-dependent antibodies or cell adhesion and migration on fibronectin (*Czuchra et al., 2006*). Moreover, contrary to the canonical model, ligand (i.e. collagen) binding to the integrins α1β1 and α2β1 has been reported to occur in the absence of integrin activation (*Abair et al., 2008a*; *Nissinen et al., 2012*). Thus, our understanding of the role of the TM interactions between α and β1 integrin subunits in regulating integrin activation and function remains incomplete.

In this work, we demonstrate that there are significant differences between the β1 and β3 integrin TM/CT structures as well as in the structure-function relationships between the αIIbβ3 and α1β1 and α2β1 integrins. These results challenge the assumption that all integrins are functionally modulated via mechanisms similar to the well-characterized αIIbβ3 integrin.

## Results

### Charge reversal of the 'snorkeling' K752 decreases α1β1- and α2β1-dependent cell adhesion and spreading on collagen

'Snorkeling' of the positively charged side chain of the membrane-buried integrin β3-K716 to form an ion pair with an anionic phosphodiester moiety of a lipid head group at the membrane-water interface is thought to regulate the integrin affinity by stabilizing the inactive state (*Kim et al., 2012*). We wished to assess whether this mechanistic component of integrin regulation is conserved in β1 integrins, specifically for the collagen binding integrins α1β1 and α2β1. Integrin β1-K752 corresponds to β3-K716 (*Figure 1A*). We therefore generated populations of integrin β1 null renal collecting duct (CD) polarized epithelial cells expressing comparable levels of human integrin β1-WT, β1-K752R and β1-K752E. The CD cells were sorted for equal levels of expression of β1 integrin and endogenous integrin α1 or α2 subunits. The different cell populations were then subjected to adhesion on collagens I or IV, which are the preferred ligands for integrins α2β1 and α1β1, respectively. The β1-K752R mutation (which maintained the positive amino acid charge) did not significantly change CD cell adhesion to collagen I (0.5 μg/ml) in the absence or presence of an integrin α1 blocking antibody (*Figure 1B*). Similar results were found over a concentration range of collagen I from 0.125–20 μg/ml (data not shown). However, under the same conditions, CD cells expressing the β1-K752E mutation (which changed the amino acid charge from positive to negative) exhibited little adhesion to collagen I in the absence or presence of an integrin α1 blocking antibody (*Figure 1B*). Comparable results were found for CD cell adhesion to collagen IV (0.25 μg/ml) (*Figure 1C*) in the presence or absence of an integrin α2 blocking antibody as well as over a concentration of collagen IV from 0.0625–10 μg/ml (data not shown). Due to the severe collagen adhesion defect of CD cells expressing the K752E-β1 mutant, we investigated the effect of this mutation on cell spreading on collagen I and collagen IV over short periods of time. CD cells expressing the β1-K752E mutation spread significantly less on collagen I than CD cells expressing wild type β1 integrin at 15, 30 and 45 min (*Figure 1D and E*). Similar results were seen when the cells were plated on collagen IV, although the difference was no longer statistically different at 45 min (*Figure 1D and F*). These results were completely contrary to expectations based on the activating nature of the K716E-β3 mutation described for αIIbβ3 (*Kim et al., 2012*).

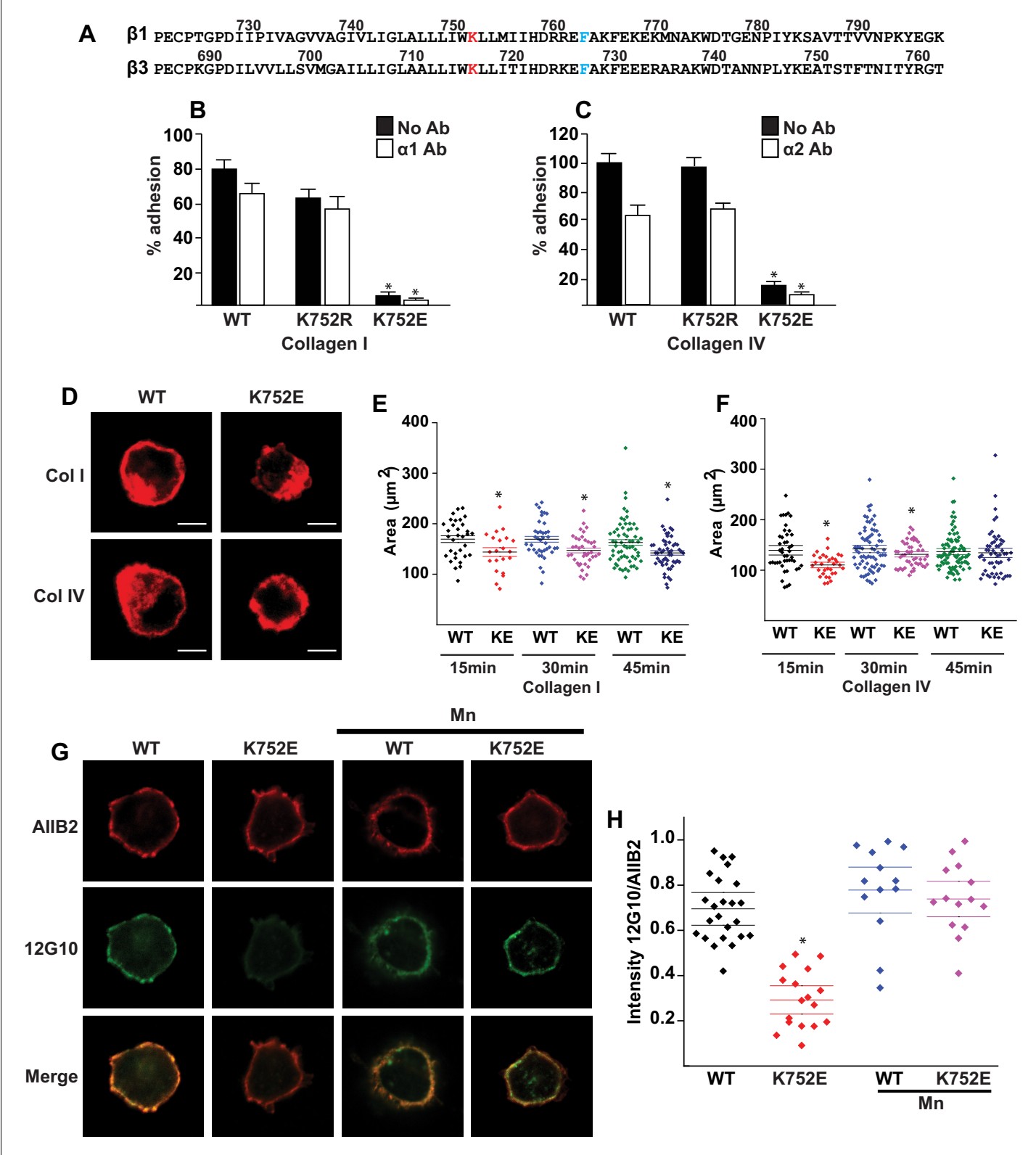

**Figure 1.** The β1-K752E mutation decreases collecting duct cell adhesion to collagens. (A) The sequences of the β1 and β3 TM/CTs are annotated. The highly conserved lysine (K752 in integrin β1and K716 in integrin β3) is colored in red. The β3 TM-only construct used in previous studies (*Kim et al., 2012*) ends at F727, which was colored in blue along with the corresponding residue in β1, F763. (B–C) The adhesion of CD cells to collagen I (0.5 µg/ml) (B) or collagen IV (0.25 µg/ml) (C) for 1 hr was measured. These assays were performed in the presence and absence of blocking antibodies to α1 or

*Figure 1 continued on next page*

*Figure 1 continued*

α2 respectively. The error bars indicate SD and * indicates a statistically significant difference (p<0.01) between WT and K752E CD cells. These assays were performed at least three times. (D–F) The spreading of wild type and K752E CD cells on collagen I and collagen IV at 15, 30 and 45 min was measured on cells stained with rhodamine phalloidin. All the images were taken close to the substrate. Representative cells at 30 min on both collagen I and IV is shown (D). The area of at least 35 cells per time point were quantified using ImageJ software and expressed graphically. Scale bar represents 10 microns (E and F). The error bars indicate SD and * indicates a statistically significant difference (p<0.05) between WT and K752E CD cells. These assays were performed at least three times. (G–H). The active conformation of integrin β1 on adherent wild type (WT) and K752E expressing CD cells that were allowed to adhere to collagen I for 1 hr was determined using 12 G10 antibody. Total integrin β1 surface expression was determined using AIIB2 antibody. All the images were taken close to the substrate. (G). The relative fluorescence intensity of 12 G10/AIIB2 as described in the Materials and methods is expressed graphically (H). The error bars indicate SD and * indicates a statistically significant difference (p<0.01) between WT and K752E CD cells. These experiments were performed at least three times each. Intensity measurements were performed on over 40 different fields for each of the samples.

We next investigated whether this unexpected major decrease in cell adhesion to the collagens by the β1-K752E mutant was due to alterations in β1 integrin activation as assessed by the amount of the β1 integrin activation epitope-reporting 12G10 antibody binding relative to the total amount of β1 integrin (as measured by AIIB2) in CD cells that adhered to collagen I. The intensity of 12G10/AIIB2 antibody binding was significantly less in the β1-K752E mutant compared to WT CD cells. This difference was no longer present when the cells were adhered in the presence of $Mn^{2+}$, which is known to artificially activate integrins (*Figure 1G and H*). Thus, reversing the basic charge of the snorkeling lysine near the TM domain of the β1 integrin significantly decreased integrin α1β1 and α2β1-dependent adhesion to and spreading on collagens by polarized CD epithelial cells. In addition, it resulted in a lower affinity state of β1 integrin in the context of CD cells adherent on collagen I.

## The β1 and β3 TM/CT have different structures

We next performed structural studies to probe the basis for the discrepancies between our results and those found when the snorkeling lysine K716 was mutated in β3. We carried out two different sets of experiments in which the buffer and bicelle model membranes were varied. The first experiment involved 0.5 mM (0.57 mol%) integrin TM/CT in q = 0.3 D6PC/DMPC bicelles and a buffer containing 250 mM imidazole, pH 6.5. DMPC/D6PC bicelles were selected as optimal relative to POPC/D6PC, DMPC/D7PC, and DMPC/cyclofos6 based on comparing the $^1$H,$^{15}$N-TROSY NMR spectra (*Figure 2—figure supplement 1*) from each composition. D6PC/DMPC bicelles yielded the most optimal β1 TM/CT spectrum in terms of the total number of peaks, evidence for conformational homogeneity, and the relative intensities of peaks from the TM to those of peaks from the cytosolic domain. The second experiment employed bicelle conditions that matched those used in critical prior studies of the isolated TM of the platelet integrin αIIb/β3 heterodimer and constituent monomers (*Kim et al., 2012*; *Lau et al., 2008a*; *Lau et al., 2009*): 0.3 mM (0.34 mol%) integrin TM/CT, 20% q = 0.3 D6PC/POPC/POPS bicelles (where POPC:POPS = 2:1 mol:mol) in a buffer composed of 25 mM HEPES, pH 7.4. For a number of key experiments (below) both sets of conditions were employed and usually yielded similar results.

TROSY-based 3-D NMR experiments were carried out to assign the backbone amide $^1$H, amide $^{15}$N, $^{13}$CA, $^{13}$CO, and (when possible) $^{13}$CB NMR resonances for the integrin β1 TM/CT in bicelles, as illustrated in *Figure 2A*. While peaks have previously been assigned for the isolated β3 TM (*Lau et al., 2009*), this is not the case for the combined TM/CT in bicelles. We therefore completed assignments for this protein as well (*Figure 2B*). For both the β1 and β3 TM/CT the assigned chemical shifts have been deposited into the BMRB database (*Ulrich et al., 2008*) (access codes: 26623 and 26624). Using the CA, CO, and CB chemical shifts for the assigned resonances, the secondary structures of both the β1 and β3 integrin TM/CT were determined using chemical shift index and TALOS-N analyses (*Figure 2C* and *Figure 2—figure supplement 2*). For β1, there is an α-helix extending from the N-terminal I732 at the beginning of the TM to K765 located in the cytosol, roughly eight sites beyond the end of the TM. For the β3 TM/CT, this helix also starts at the beginning of the TM (at I693), but extends roughly 16 residues into the cytosol, terminating only at A737. Beyond the TM/CT helix, the remaining intracellular domains of both integrin β1 and integrin β3 are

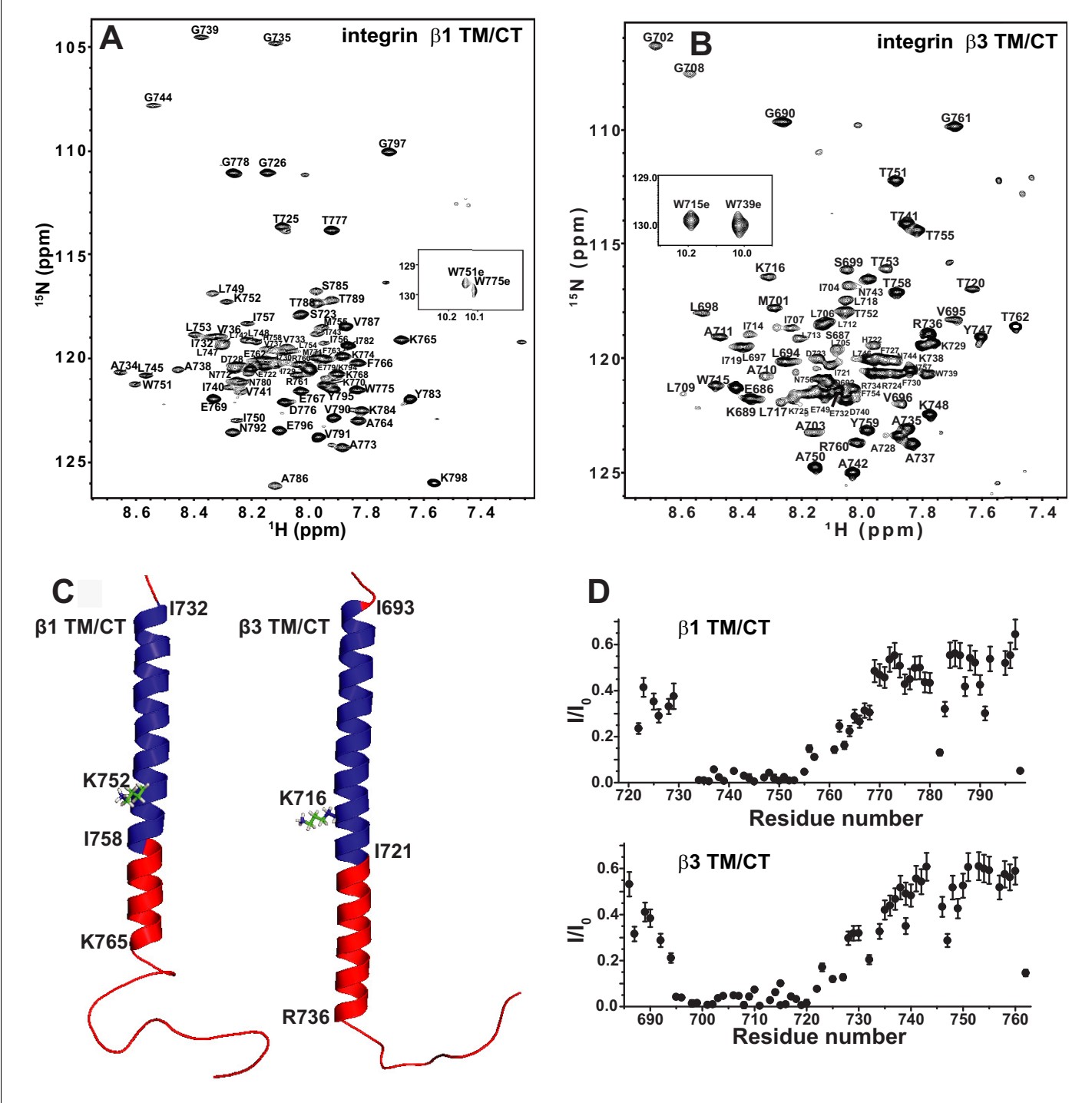

**Figure 2.** The β1 and β3 transmembrane and cytosolic domains have distinct structures. (**A**) 800 MHz $^1$H-$^{15}$N TROSY spectrum of the WT integrin β1 TM/CT with peak assignments shown. This spectrum was collected at 45°C and the sample contains 500 μM β1 TM/CT (0.57 mol%), 20% q = 0.3 D6PC-DMPC bicelles, 1 mM EDTA, 250 mM IMD at pH 6.5% and 10% $D_2O$. (**B**) 900MHz $^1$H-$^{15}$N TROSY spectrum of the WT integrin β3 TM/CT with peak assignments shown. This spectrum was collected at 45°C and the sample contained 300 μM β3 TM/CT, 20% q = 0.3 D6PC-DMPC bicelles, 1 mM EDTA, 250 mM IMD buffer at pH 6.5, with 10% $D_2O$. (**C**) Structural comparison of integrin β1 and β3 TM/CT in D6PC/DMPC bicelles. These structural models are based on backbone dihedral angles as determined by TALOS-N and chemical shift index analysis of backbone NMR chemical shift values. The figure was made in PYMOL. As described in the results section, the CT segments of the helices observed for both β integrins appear to be subject to significant dynamic fraying. (**D**) Assessment of exchange of protons between backbone amide sites and water at 100 msec, as determined by the
*Figure 2 continued on next page*

*Figure 2 continued*

CLEANEX-PM NMR experiment. Sites with low peak intensities after the 100 msec mixing period relative to control conditions ($I/I_0$) are resistant to hydrogen exchange, while sites with high $I/I_0$ values exchange rapidly on the time scale of 100 msec.

The following figure supplements are available for figure 2:

**Figure supplement 1.** 900 MHz $^1H$,$^{15}N$-TROSY spectra of integrin β1 TM/CT.

**Figure supplement 2.** NMR chemical shifts to determine alpha helical content of the integrin β1 and β3 TM/CT.

unstructured. The degree to which the TM helix of the β3 integrin extends further into the CT relative to β1 (nine residues longer for β3) is striking. The stability of this helix was probed by examining backbone amide/water hydrogen exchange rates using the CLEANEX-PM NMR experiment (*Hwang et al., 1998*) and by measuring backbone amide $^{15}N$ NMR relaxation rates. Results from these experiments (*Figure 2D* and *Figure 2—figure supplement 2*) show that the transmembrane helix is rigid and, not surprisingly, exchange-resistant. The CT segments identified as helical by analysis of the chemical shifts are seen to exhibit a gradient of low exchange/low motion to significantly higher exchange and motion going from the end of the TM to the end of the chemical-shift identified CT helical segments. There is also a modest spike in exchange and motion observed at the TM/CT interface. These data are consistent both with the notion that the CT helices are prone to significant fraying and also that there is a modest degree of hinge motion at the TM/CT helix interface. It is very clear from the topological analysis presented in the following sections that the CT helix is extended away from the membrane into the cytosol and that any hinge motions at the TM/CT interface are not of sufficient magnitude to enable significant interactions of the CT helix with the membrane surface.

## The charge of the snorkeling lysine in the β1 and β3 integrin TM domains is dispensable for defining the TM membrane topology

The membrane topologies of the β1 and β3 TM/CT were probed using NMR spectroscopy in the presence of hydrophilic (Gd(III)-DTPA) and hydrophobic (16-DSA) paramagnetic probes. Probe accessibility to backbone amide 1H sites was assessed as the degree of site-specific backbone amide TROSY peak broadening due to the proximity of the paramagnetic probe, which was quantitated as the ratio between the peak intensity in a probe-containing sample versus the corresponding peak intensity from a matched diamagnetic (probe-free) control sample. A low ratio corresponds to high probe access.

We first investigated the membrane topology of WT integrin β1 TM/CT in DMPC/D6PC bicelles at pH 6.5 by comparing peak intensities (*Figure 3*) in NMR spectra acquired in the absence (black peaks) and presence (red peaks) of either water soluble Gd(III)-DTPA or lipophilic 16-DSA paramagnetic probes. The plateau in peak ratios observed when Gd(III)-DTPA was used, combined with the corresponding trough in the ratio when 16-DSA was used indicates that the beginning and ends of the TM are near sites I732 and I757, respectively. Both the extracellular segment and the cytosolic domains are largely exposed to the hydrophilic Gd(III)-DTPA (*Figure 3A and C*), although the modest degree of protection to Gd(III)-DTPA near the $P_{781}IY$ segment (see spike in right side of *Figure 3C*) suggests that this segment undergoes transient interactions with the bicelle surface. Importantly, these results show that, similar to integrin β3, the β1 TM includes both the putative 'snorkeling' Lys site (752 in β1, 716 in β3) and the five hydrophobic residues following it. The β1 TM is terminated by the $HDRRE_{762}$ motif, where D759 corresponds to D723 in β3, which forms a critical salt bridge with R995 in the αIIb subunit as part of the 'inner clasp' that stabilizes the inactive integrin heterodimeric state (*Lau et al., 2009*).

Probe accessibility measurements were repeated for the TM/CT of both the β1-K752E and β1-K752R mutants. Surprisingly, both mutants yielded almost exactly the same patterns of accessibility to both paramagnetic probes as WT (*Figure 3A and C*, *Figure 3—figure supplement 1*), indicating that both charge-conservative and charge reversal mutations at K752 induce little change in the membrane topology or tilt of the TM helix of β1 TM/CT. The same results were obtained when the

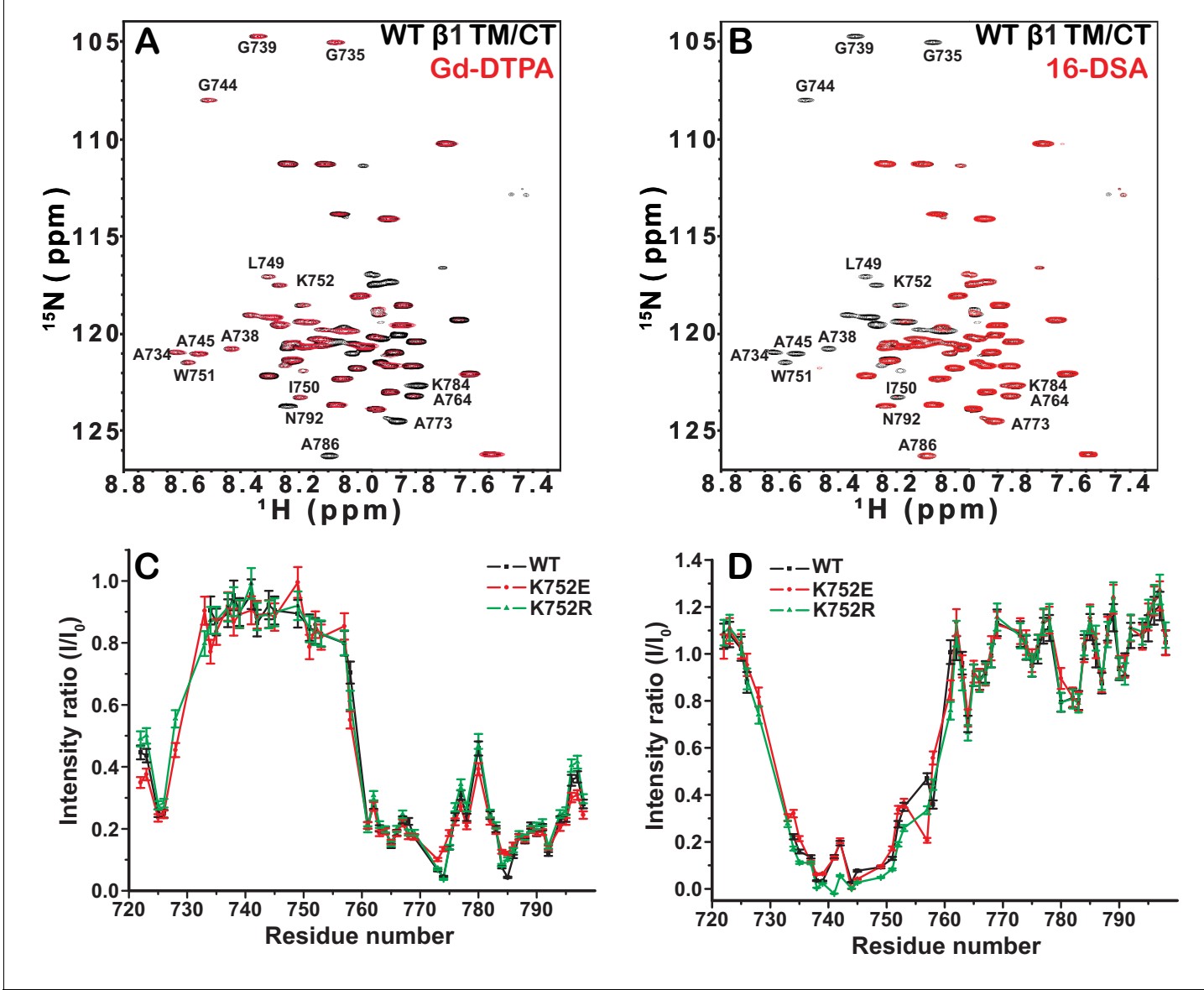

**Figure 3.** Examination of the bilayer topology of integrin WT β1 TM/CT and two mutants (K752E, K752R) in bicelles using NMR and paramagnetic probes. Peak intensity changes in the 600MHz [1]H-[15]N TROSY spectra are reported for the integrin WT β1 TM/CT and two mutants (K752E, K752R) as induced either by 4 mol% 16-DSA as the hydrophobic paramagnet or by 10 mM Gd-DTPA as the water soluble paramagnet. The spectra were collected at 45°C and the samples contained ~500 μM β1 TM/CT, 20% q = 0.3 D6PC-DMPC bicelles, 1 mM EDTA, 250 mM IMD buffer at pH 6.5, with10% $D_2O$. The overlaid spectra for WT β1 TM/CT containing no paramagnetic probe (black, bottom) and with Gd-DTPA (red, on top) are shown (**A**), while the corresponding spectra for the two mutants are shown in supporting *Figure 3*. The absence of a red peak indicates that the peak is broadened beyond detection by the presence of the paramagnetic probe. The overlay of spectra for WT β1 TM/CT containing 16-DSA (red) versus no paramagnetic probe (black) are shown (**B**). Panel **C** shows the peak intensity changes induced by 10 mM Gd-DTPA on the three forms of the integrin β1 TM/CT, while panel **D** shows the results for 16-DSA. The intensity ratio ($I/I_0$) reported for each TROSY amide resonance is the observed peak intensity in the presence of the paramagnetic probe divided by the intensity of the corresponding peak under probe-free diamagnetic conditions. Two replicates of this experiment giving consistent results were performed. We show one representative example. The error is estimated from the noise level, or 5% of value, whichever is larger.

The following figure supplement is available for figure 3:

**Figure supplement 1.** [1]H,[15]N-TROSY spectral overlays of integrin β1 TM/CT in D6PC/DMPC bicelle with no paramagnetic probe (black) and with either 10 mM Gd-DTPA (red) or 4 mol% 16-DSA (red).

lipophilic paramagnetic probe 16-DSA was used (*Figure 3B and D*, *Figure 3—figure supplement 1*).

The fact that the β1-K752E mutation did not impact the membrane topology/tilt for the β1 subunit was surprising in light of a previous study that presented Mn(II)-EDDA (polar) probe accessibility data showing that the β3-K716E mutation induced a major change in membrane topology, at least for the isolated integrin β3 TM (*Kim et al., 2012*). Specifically, a pronounced tilt of the TM for the WT β3 protein was reported to be largely reversed by the K->E mutation to yield a TM that is much more closely aligned with the bilayer normal. It was proposed in that work that the tilt of the β3 TM is stabilized by a 'snorkeling' interaction between the side chain amino moiety of the membrane-buried K716 site and oxyanions in the nearby lipid head groups (*Kim et al., 2012*). Because of the surprising discord between the β1 results of this work versus the prior β3 results we repeated the Mn (II)-EDDA probe experiment for the isolated β3 TM using the same methods as originally used by Kim, Schmidt, et al (*Kim et al., 2012*). As shown in *Figure 4A* and *Figure 4—figure supplement 1*, our β3-WT vs. β3-K716E results are very similar to the previously reported results. Use of the hydrophilic Gd-DTPA probe led to a similar result as for Mn-EDDA (*Figure 4B* and *Figure 4—figure supplement 1*), yielding results consistent with a major change in tilt for the TM domain due to the β3-K716E mutation. However, a discordant result was obtained when the lipophilic 16-DSA was used as the paramagnetic probe (note that an apolar probe was not employed in the previous study (*Kim et al., 2012*) of the isolated β3 TM): for 16-DSA, little difference was observed between the membrane topology and tilt of β3-WT and the β3-K716E mutant (*Figure 4C* and *Figure 4—figure supplement 1*). This led us to recall literature showing that chelate probes such as Mn(II)-EDDA and Gd(III)-DTPA sometimes have an exposed ligand site that can transiently associate with anions such as Glu and Asp side chain carboxylates (*Hocking et al., 2013*). To test this possibility, we repeated the Gd(III)-DTPA probe experiment under conditions in which excess (10 mM) free EDTA was added to the solution to cap any free ligand sites in Gd(III)-DTPA, thereby suppressing any direct binding of protein carboxyl sides to the open ligand site in the lanthanide ion chelate. Under these conditions the β3-WT and β3-K716E mutant exhibited the same probe accessibility patterns (*Figure 4D* and *Figure 4—figure supplement 1*), fully consistent with the 16-DSA results. These results indicate that the previously reported differences seen for the isolated β3 TM using Mn(II)-EDDA reflected an experimental artifact based on the tendency of free carboxyl groups in proteins (i.e., the introduced Glu716 side chain carboxyl) to transiently serve as ligands to available metal ion coordination sites in chelate complexes.

We used both additional paramagnetic probe experiments and backbone amide $^{15}$N NMR R2 relaxation rate measurements to confirm that the similarity of the membrane topology/tilt of β3-WT vs. β3-K716E is maintained when the full cytosolic domain is included (TM/CT, *Figure 5A–5D* and *Figure 5—figure supplement 1*). We also verified that these results were not altered by major changes in bicelle or buffer composition, pH, or temperature (*Figure 5A and B* vs. 5C and 5D, also *Figure 5—figure supplement 2*). Finally, the data of *Figure 5E and F* and *Figure 5—figure supplements 2* and *3* complements that of *Figure 3C and D* by showing that membrane tilt/topology for both β1-WT and β1-K752E TM/CT does not strongly depend on lipid composition, buffer type, pH, or temperature.

These results indicate that for both the β1 and β3 integrins, the lysine buried in the membrane at the membrane/cytosol interface does not play a major role in dictating its TM topology or tilt, at least not for the free β subunits. This finding does not rule out the possibility of snorkeling ion pairing of the β3-K716 and β1-K752 side chains with the lipid head group, but does rule out the notion that such interactions play a decisive role in determining TM helix tilt.

## Impact of K716E/K752E mutations on β-subunit TM/CT heterodimerization with α subunits

The β3-K716E mutation results in constitutive activation of the αIIbβ3 integrin and α5β1 integrins (*Kim et al., 2012*). To quantitate the impact of the β3-K716E and β1-K752E mutations on heterodimer formation NMR was used to monitor titrations in which WT and mutant forms of $^{15}$N-labeled integrin β1 and β3 TM/CT were titrated by various unlabeled WT α TM/CT subunits (*Figure 6A–D* and *Figure 6—figure supplement 1*). Binding of integrin α5 to either the WT or K752E β1 TM/CT and binding of integrin αIIb to either WT or K716E β3 resulted in the disappearance of the β subunit TROSY NMR peaks rather than shifts, indicating 'slow exchange' between free and complexed

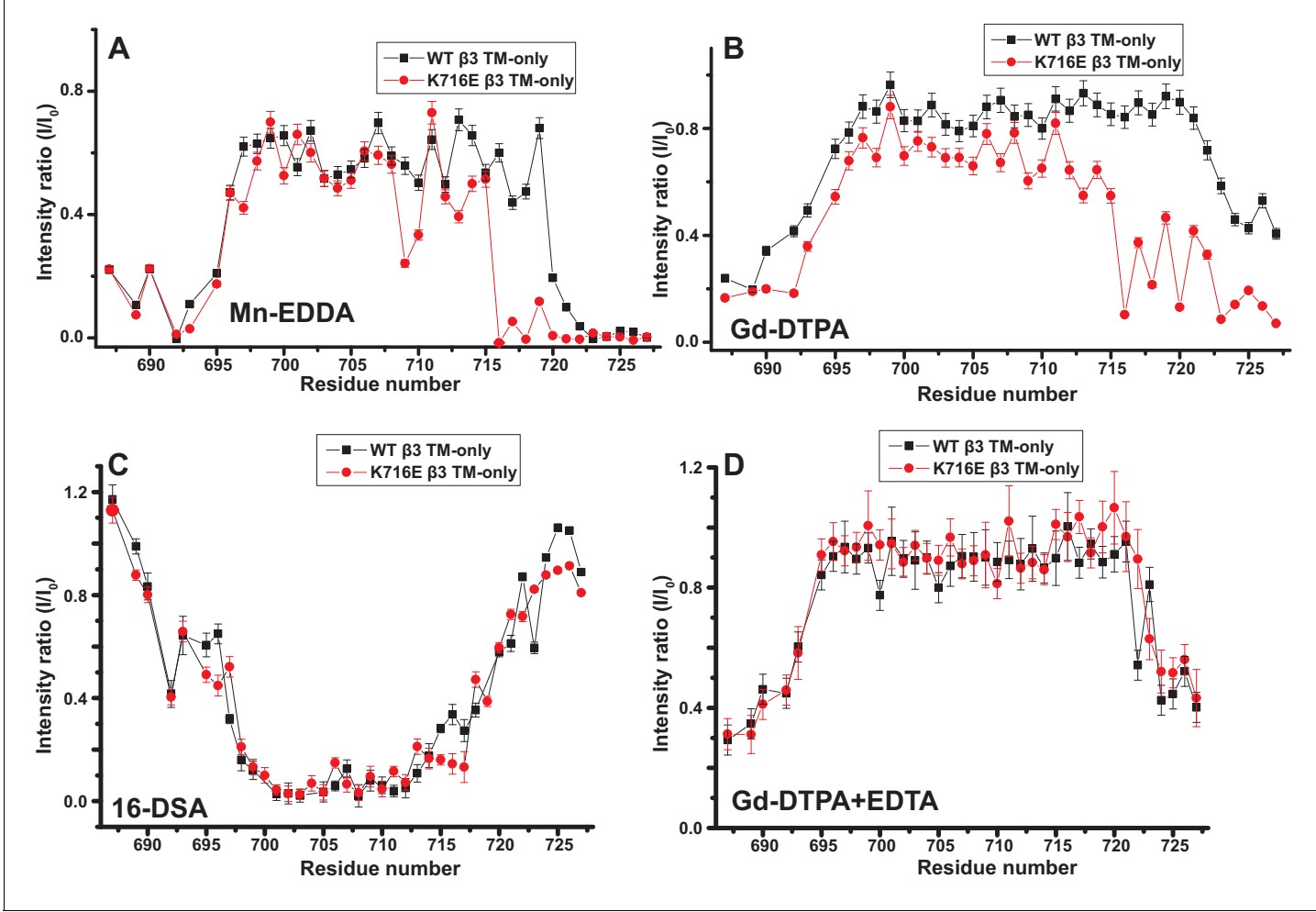

**Figure 4.** Examination of the bilayer topology of isolated WT integrin β3 TM and of β3 K716E TM in bicelles using NMR and paramagnetic probes. Paramagnetic probe-induced intensity changes are reported for the peaks in 900 MHz $^1$H-$^{15}$N TROSY spectra of WT integrin β3 TM and β3 K716E TM at 45°C: (A) 1 mM Mn-EDDA, (B) 10 mM Gd-DTPA, (C) 10 mM Gd-DTPA plus 10 mM EDTA, (D) 4 mol % 16-DSA. The NMR samples contained 0.3 mM protein, 20% D6PC/POPC/POPS bicelles (q = 0.3, 2:1 POPC:POPS), 25 mM HEPES at pH 7.4, with 10% $D_2O$. The intensity ratio ($I/I_0$) reported for each TROSY amide resonance is the observed peak intensity in the presence of the paramagnetic probe divided by the intensity of the corresponding peak under probe-free diamagnetic conditions. Three replicates of these experiments giving consistent results were performed. We show one representative example. The error is estimated from noise level, or 5% of value, whichever is larger.

The following figure supplement is available for figure 4:

**Figure supplement 1.** 900 MHz $^1$H,$^{15}$N-TROSY spectral overlays of WT integrin β3 TM in D6PC/POPC/POPS (2:1) bicelles with no paramagnetic probe (black) and with either 10 mM Gd-DTPA (red) (A), 1 mM Mn-EDDA (red) (B), both 10 mM Gd-DTPA and 10 mM EDTA (red) (E), or with 4 mol% 16-DSA (red) (F).

subunits on the NMR time scale (*Figure 6—figure supplement 1*). For each of these titrations a 1:1 binding model was fitted to the concentration dependence of the disappearance of multiple peaks using a global fitting, in which the traces from multiple peaks were simultaneously fit. It can be seen that this approach generated reasonably good fits of the data (*Figure 6A–D*). However, traces for a few peaks, such as that of G744 in *Figure 6B*, exhibited deviation from ideal 1:1 behavior, most likely because the presence of a second concentration-dependent phenomenon such as non-specific interactions between subunits that also contributed to the observed shifts for such peaks. Eliminating the outliers from the global fitting process did not dramatically change the Kd estimated from

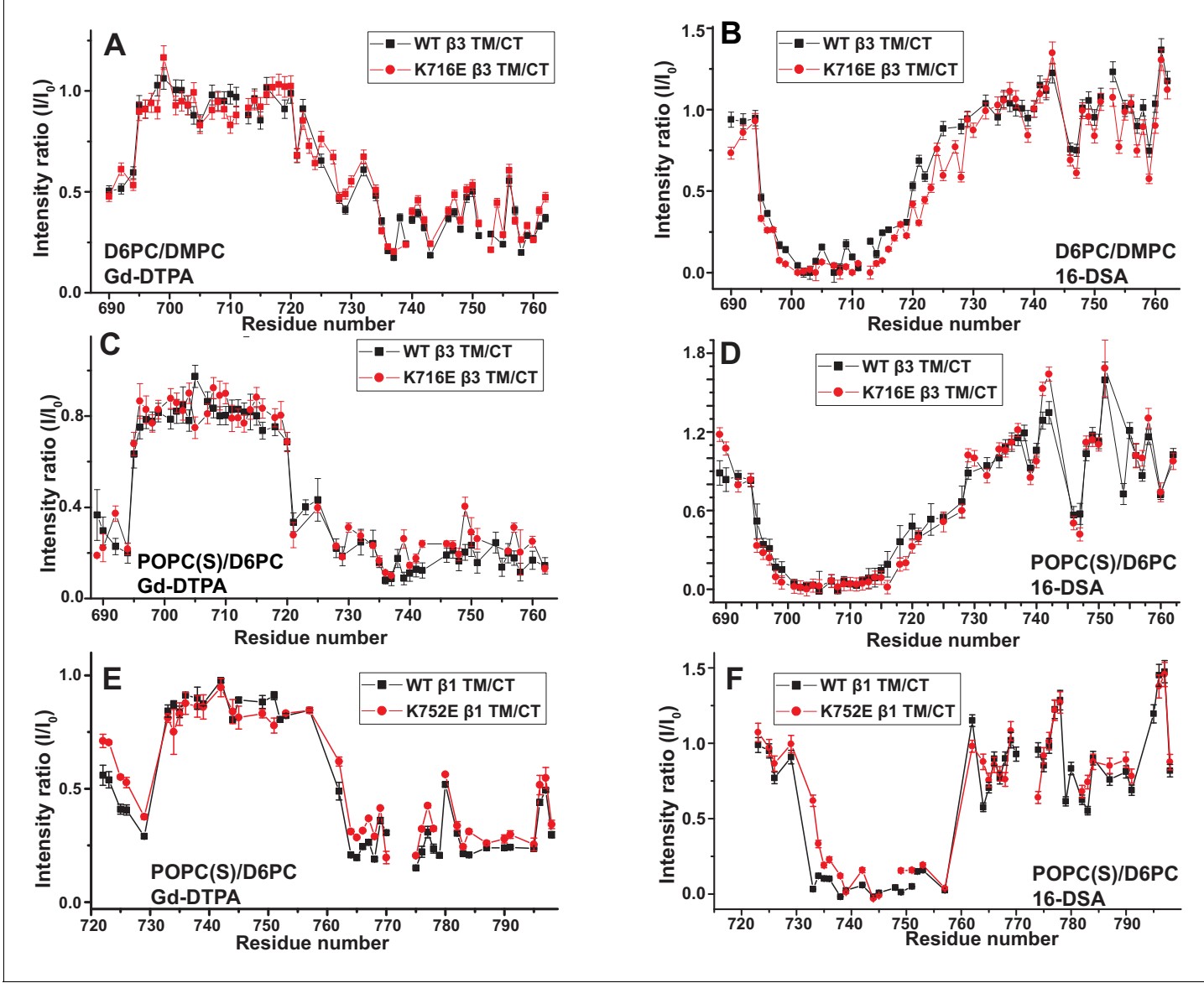

**Figure 5.** Examination of the bilayer topology of WT integrins in bicelles using NMR and paramagnetic probes. Peak intensity changes are reported as induced by 10 mM Gd-DTPA (**A**) or 4% 16-DSA (**B**) in the 900MHz $^1$H-$^{15}$N TROSY spectra of WT integrin β3 TM/CT and of the K716E mutant at 45°C in 20% q = 0.3 D6PC/DMPC bicelles, 1 mM EDTA, 250 mM IMD pH 6.5, with 10% D$_2$O. (**C**) and (**D**) are the corresponding plots for integrin β3 TM/CT and for the K716E β3 TM/CT mutant in 20% q = 0.3 D6PC/POPC/POPS bicelles, 25 mM HEPES pH 7.4. In these cases 10 mM EDTA was also included to sequester any free Gd$^{3+}$ and to cap any open metal ion ligand sites of the Gd-DTPA complex. (**E**) and (**F**) are the corresponding plots for WT integrin β1 TM/CT and for the K752E mutant form in 20% q = 0.3 D6PC/POPC/POPS bicelles, 1 mM EDTA, 25 mM HEPES pH 7.4, with 10% D$_2$O. The intensity ratio (I/I$_0$) reported for each TROSY amide resonance is the observed peak intensity in the presence of the paramagnetic probe divided by the intensity of the corresponding peak under probe-free diamagnetic conditions. At least two replicates of these experiments giving consistent results were performed. We show representative examples. The error is estimated from the noise level, or 5% of value, whichever is larger.

The following figure supplements are available for figure 5:

**Figure supplement 1.** TROSY spectra of β3 TM/CT with and without paramagentic probes.

**Figure supplement 2.** TROSY spectra of β3 TM/CT with and without paramagentic probes.

**Figure supplement 3.** TROSY spectra of of β1 TM/CT with and without paramagentic probes.

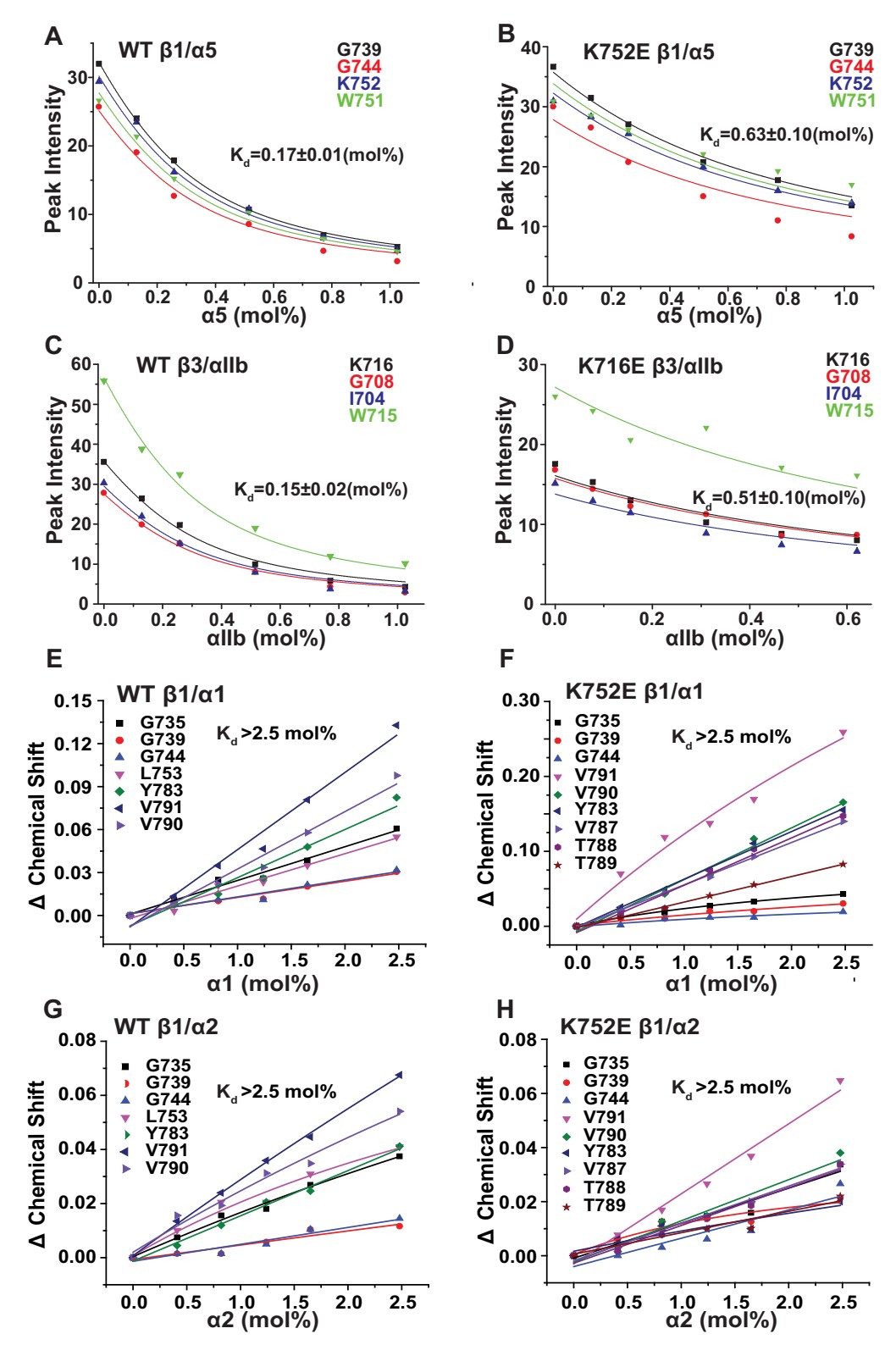

**Figure 6.** Fits of at 1:1 binding model to the data from NMR-monitored titrations of wild type and mutant $^{15}$N-labeled integrin β TM/CT subunits by unlabeled wild type α TM/CT subunits, with the best fit $K_d$ determined in each case as shown. The residue assignments for the TROSY peaks used for these analyses are indicated. In some cases (panels E-H) the binding was so weak that it was only possible to determine a lower limit to the $K_d$.

*Figure 6 continued on next page*

*Figure 6 continued*

Measurements were carried out at 45°C in 20% (w/v) D6PC/POPC/POPS (POPC:POPS = 2:1), q = 0.3, 50 mM phosphate buffer with 1 mM EDTA in 10% D$_2$O, pH 6.5.

The following figure supplements are available for figure 6:

**Figure supplement 1.** Superimposed 900 MHz $^1$H-$^{15}$N-TROSY spectra from titrations of $^{15}$N-labeled integrin β TM/CTs with unlabeled α subunit TM/CTs: α5β1(WT), α5β1(K752E) αIIbβ3(WT), and αIIbβ3(K716E).

**Figure supplement 2.** Superimposed 600 MHz $^1$H-$^{15}$N-TROSY spectra from titrations of $^{15}$N-integrin β TM/CTs with unlabeled α subunit TM/CTs: α1β1 (WT), α2β1(WT), α1β1(K752E), and α2K752Eβ1(K752E).

the fitting. For example, global fitting of all four traces shown in *Figure 6B* led to a Kd of 0.63 ± 0.10 mol%, while a repeat fit minus the G744 outlier data led to a Kd of 0.73 ± 0.1 mol%.

Because the β1 integrin subunit pairs with many other α subunits besides α5 we tested for the generality of the above results for α5β1 by examining binding of other α subunit TM/CT, namely α1 and α2. NMR titrations of wild type and mutant β1 TM/CT with the wild type α1 and α2 subunits were carried out and it was observed that the β1 peaks shifted during the titration instead of disappearing (*Figure 6—figure supplement 2*), indicative of rapid exchange on the NMR time between free and complexed species. Global fitting a 1:1 binding model to the concentration dependence of peaks seen to shift the most in each titration yielded good fits (*Figure 6E–H*) but revealed that binding did not approach saturation over the concentration range accessible by NMR, such that only lower limits for the Kd could be determined. It is interesting that, for a given concentration, titration with α1 TM/CT generated significantly larger resonance shifts in certain peaks (*Figure 6E and F*) than α2 (*Figure 6G and H*). It is also notable the largest shifts seen in the titration of K752E β1 TM/CT by α1 (*Figure 6F*) are significantly larger than the corresponding α1-induced shifts seen for the WT β1 TM/CT. Finally, it was surprising that the largest shifts seen in titrations of both WT and mutant forms of β1 are for peaks from sites on the distal cytosolic end of the CT. However, the exact interpretation of these observations and whether they are truly significant in terms informing on

**Table 1.** Dissociation constant (Kd) of integrin β1 and β3 subunits (WT and K752E mutant) with α subunits in D6PC/POPC/POPS (POPC:POPS=2:1) q=0.3 bicelles.

| Titration | Kd (mol%) | Monitored subunit | Method |
|---|---|---|---|
| α5β1 | 0.17±0.1 | 15N-β1 | NMR |
| α5β1 | 0.07±0.02 | α5 | Anisotropy |
| **α5β1KE** | 0.63±0.1 | 15N-β1-KE | NMR |
| α5**β1KE** | 0.80±0.27 | α5 | Anisotropy |
| αIIbβ3 | 0.15±0.1 | 15N-β3 | NMR |
| αIIb**β3** | 0.09±0.03 | αIIb | Anisotropy |
| αIIb**β3KE** | 0.51±0.1 | 15N-β3-KE | NMR |
| αIIb**β3KE** | 0.33±0.05 | αIIb | Anisotropy |
| **α1**β1 | >2.48 | 15N-β1 | NMR |
| α1**β1** | >3.2 | α1 | Anisotropy |
| **α1**β1KE | >2.48 | 15N-β1-KE | NMR |
| α1β1KE | >3.2 | α1 | Anisotropy |
| **α2**β1 | >2.48 | 15N-β1 | NMR |
| α2β1 | >3.2 | α2 | Anisotropy |
| **α2**β1KE | >2.48 | 15N-β1-KE | NMR |
| **α2**β1KE | >3.2 | α2 | Anisotropy |

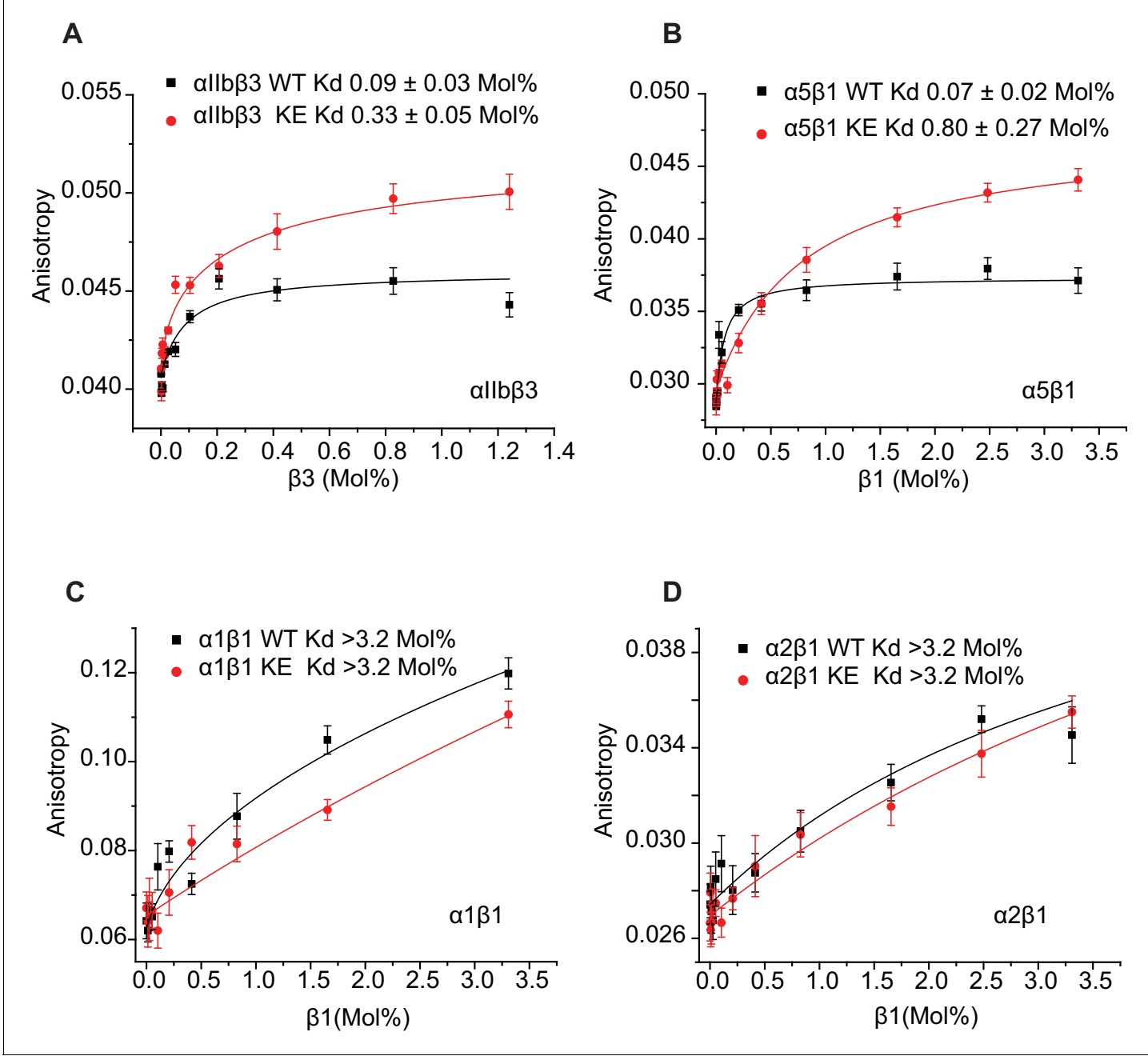

**Figure 7.** Use of fluorescence anisotropy to determine $K_d$ for complex formation between $\alpha$ and $\beta$ integrin TM/CTs. Measurements were carried out in D6PC/POPC/POPS bicelles (2% total amphiphile, q = 0.3) in 25 mM HEPES buffer pH 7.4 at 35°C. $\alpha$ subunits were labeled with IAEDAN and titrated by unlabeled $\beta$ subunits. Dissociation constants were reported in mol% of total protein to total lipid present in the bicelles for: (A) $\alpha$IIb$\beta$3, (B) $\alpha$5$\beta$1, (C) $\alpha$1$\beta$1, (D) $\alpha$2$\beta$1. At least three independent experiments were carried out. Shown is an example of each experiment where at least five measurements were made at each time point. The error bars depict the standard deviation of these five measurements.

integrin activation is difficult to assess in light of the fact that heterodimerization in all four cases (*Figure 6E–H*) was too weak to quantify.

That the NMR titration results of *Figure 6* report primarily on the formation of stoichiometric complex formation between $\alpha$ and $\beta$ TM/CT subunits is supported by the results of corresponding fluorescent anisotropy titrations in which fluorescently-labeled $\alpha$ TM/CT subunits were titrated by unlabeled $\beta$ TM/CT subunits, followed by fitting of the data by a 1:1 binding model (*Figure 7*). The

various $K_d$ determined by both NMR and fluorescence methods are presented in *Table 1*, where it is seen that for each subunit combination the NMR titrations conducted at pH 6.5°C and 45°C and fluorescence anisotropy titrations conducted at pH 7.4°C and 35°C yielded similar results.

The binding results of *Table 1* show that, as expected, the αIIb TM/CT forms a relatively avid complex with the β3 TM/CT (Kd of ca. 0.1 mol%) (*Table 1*). Also as expected, the β3-K716E mutation reduced the affinity of this complex by roughly 4-fold (to a $K_d$ of ca. 0.4 mol%). Binding of WT α5 TM/CT to β1 TM/CT was also seen to be avid ($K_d$ of ca. 0.1 mol%) (*Table 1*), while the K752E mutation in the β1 TM/CT reduces its binding affinity for α5 by a factor of 7 ($K_d$ of ca. 0.7 mol%). This is consistent with a previous study that showed the β1-K752E mutation induces constitutive activation of the α5β1 integrin (*Kim et al., 2012*).

It was very surprising that for all 4 combinations of α/β association of β1-WT and β1-K752E TM/CT with the α1 and α2 TM/CT that binding was so weak it could in no case be quantitated (Kd > 3.2 mol% for both α1β1 and α2β1 TM/CT) (*Table 1*).

Together these data suggest that the TM-K>E mutation decreases the affinity of the αβ heterodimers of both αIIbβ3 and α5β1 resulting in increased 'activity' of these integrins. These results also suggest that full length α1β1 and α2β1 integrins may be 'constitutively active' under physiological conditions and cannot be further activated.

## Cell adhesion does not correlate with alterations of α and β integrin TM/CT affinity

Due to these major discrepancies in the biophysical behavior of α5β1 and the collagen-binding integrins α1β1 and α2β1, we assessed whether integrin α5β1-dependent adhesion and spreading on fibronectin was increased in response to the introduction of the β1-K752E mutation. CD cells with equal surface expression of the α5, β1 and β1-K752E mutation were generated. Both the wild type and β1K752E CD cells adhered equally to fibronectin (0.5 µg/ml) (*Figure 8A*). As these cells express integrin αv, the same experiment was carried out in the presence of an integrin αv blocking antibody. Under these conditions CD cell adhesion was decreased, albeit less than the adhesion defect observed with the CD cells adhering to collagen I or IV. No differences in cell adhesion were seen when the cells adhered to fibronectin in the presence of an anti-β1 antibody, suggesting that the β1-K752E mutation decreased integrin α5β1-dependent adhesion of CD cells to fibronectin. When we assessed the spreading of these cells on fibronectin over different time periods, we observed a consistent but non-significant decrease in spreading of the β1K752E CD cells at 15-, 30- and 45 min time points (*Figure 8B*). We also assessed the relative intensity of 12 G10 compared to AIIB2 on cells plated on fibronectin. Similar to when the cells adhered to collagen I there was decreased intensity of 12 G10 (*Figure 8C and D*), which correlated with the adhesion and spreading defects. Thus a β1-K752E mutation decreases CD cell adhesion, spreading and integrin α5β1 activation when CD cells bind to fibronectin.

## Impact of the β1-K752E and β3-K716E mutations on talin-F3 binding

The F3 domain of talin contains a phosphotyrosine-binding (PTB) signature that binds to the NPxY motif shared by the CT of both integrins β1 and β3, while the linked F2 domain enhances binding through favorable electrostatic interactions with anionic lipids in the vicinity of the integrin on the membrane surface (*Moore et al., 2012*; *Saltel et al., 2009*). We probed whether affinity of the interactions of the isolated talin F3 domain with the β1 and β3 TM/CT are altered by the β1-K752E and β3-K716E mutations. NMR was used to monitor titrations of $^{15}$N-labeled integrin TM/CT subunits with unlabeled talin F3 in bicelles containing POPS, an anionic lipid (*Figure 9*). During the course of the titrations it was observed that certain peaks disappeared, indicating that binding between the two proteins is slow on the NMR time scale, as is typical when Kd $\leq$ 100 µM. Binding was therefore quantitated based on monitoring the F3 domain concentration-dependence of peak intensity reductions. As shown in *Figure 9*, the talin F3 bound to both β1-WT and β1-K752E TM/CT in POPC/POPS/D6PC bicelles with similar (moderate) affinity ($K_d$ of 27 ± 4 µM and 18 ± 3 µM for WT and K752E β1 TM/CT, respectively). In contrast, the β3-K716E mutation results in a 6-fold increase in the $K_d$ for talin-1 binding from 5 ± 2 µM to 30 ± 8 µM, suggesting the β3-K716E mutation alters the structure and/or dynamics of bicelle-associated β3 TM/CT to reduce talin-1 F3 binding affinity (*Figure 10*). These results indicate that mutation of the snorkeling lysine of β3, but not of β1, significantly

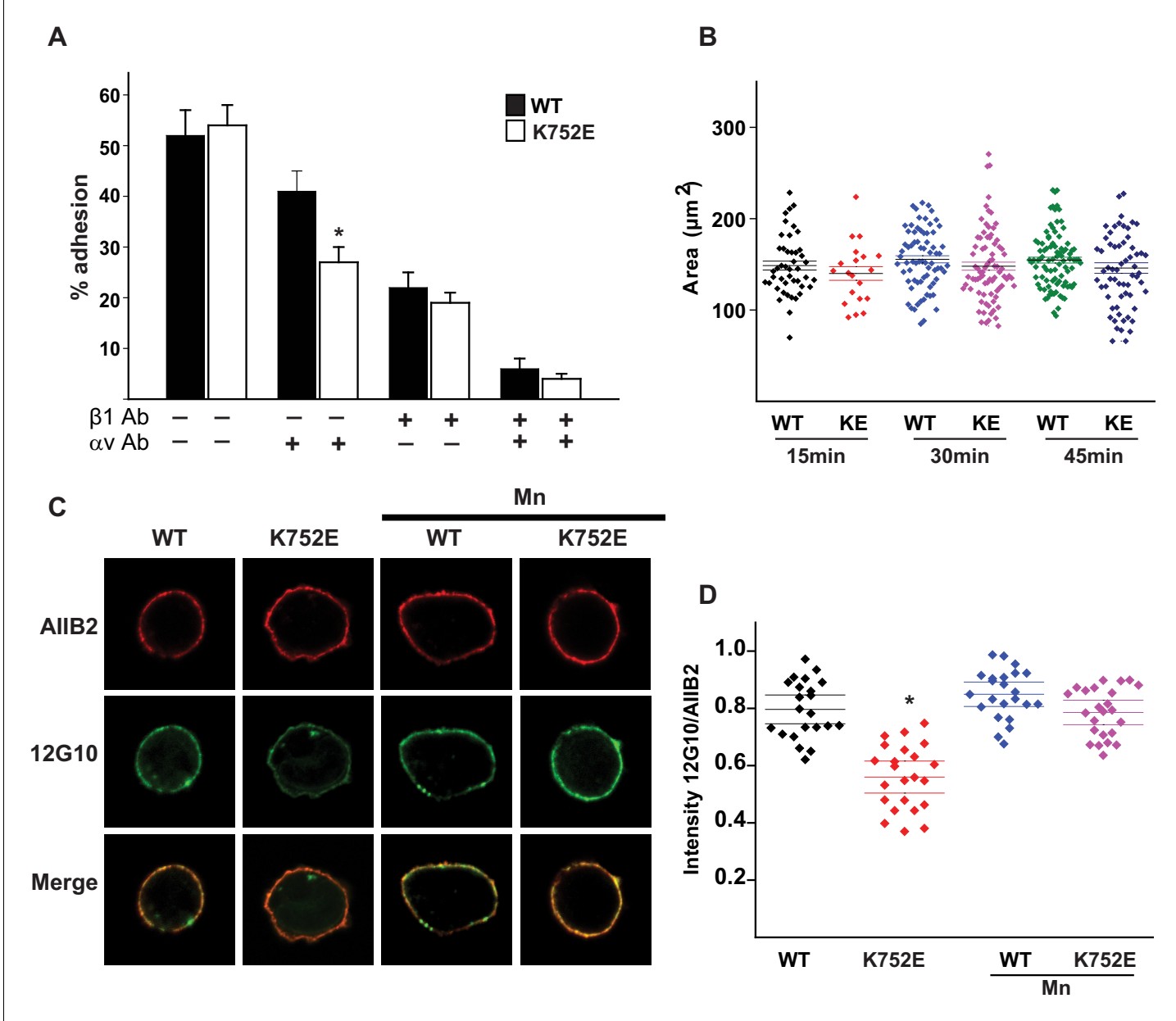

**Figure 8.** The β1-K752E mutation decreases collecting duct cell adhesion to fibronectin. (**A**) The adhesion of CD cells to fibronectin (0.5 μg/ml) for 1 hr was measured. These assays were carried out in the presence or absence of blocking antibodies directed against the αv or β1 subunits. The error bars indicate SD. * indicates a statistically significant difference (p<0.01) between WT and K752E CD cells. These assays were performed at least three times. (**B**) The spreading of wild type and K752E CD cells on fibronectin at 15, 30 and 45 min was measured on cells stained with rhodamine phalloidin. The area of at least 35 cells per time point were quantified using ImageJ software and expressed graphically. The error bars indicate SD. These assays were performed at least three times. (**C–D**) The active conformation of integrin β1 on adherent wild type (WT) and K752E expressing CD cells that were allowed to adhere to fibronectin for 1 hr was determined using 12 G10 antibody. Total integrin β1surface expression was determined using AIIB2 antibody (**C**). (**D**) The fluorescence intensity was measured as described in the Materials and methods. The error bars indicate SD and * indicates a statistically significant difference (p<0.01) between WT and K752E CD cells. These experiments were performed at least three times each. Intensity measurements were performed on over 40 different fields for each of the samples.

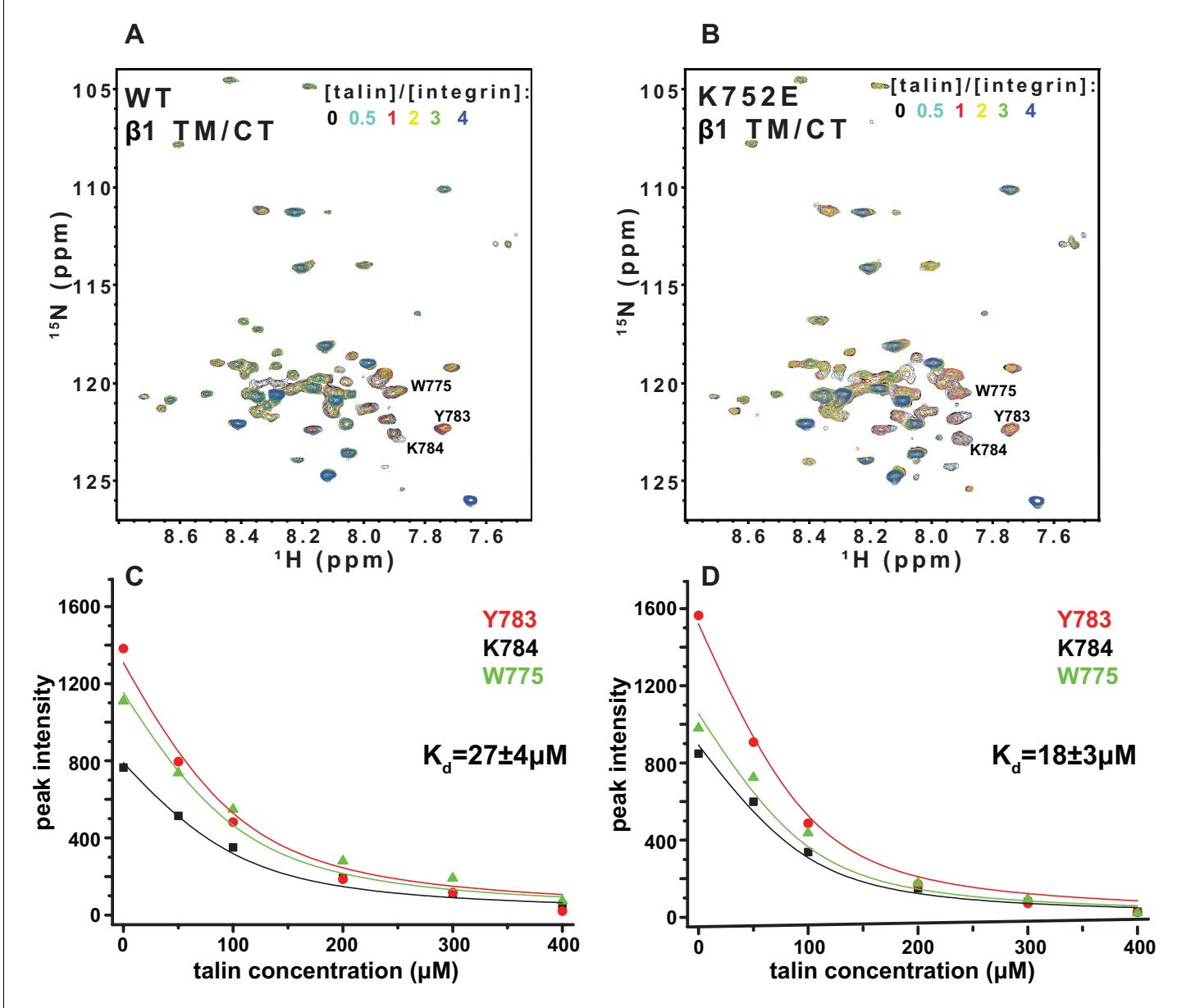

**Figure 9.** NMR-monitored titrations of integrin β1 TM/CT by the talin-F3 domain. $^{15}$N-labeled WT and K752E mutant integrin β1 TM/CT samples were titrated with unlabeled talin1-F3. The concentration of all integrins was 100 μM and the samples contained 5% q = 0.3 D6PC/POPC/POPS (POPC: POPS = 2:1) bicelles in 25 mM HEPES buffer at pH 7.4. The $^{1}$H,$^{15}$N-TROSY NMR spectra were collected at 35℃ at 900MHz (upper panels). In each case six titration points were collected in which the mole ratio between talin1-F3 and the integrin was varied: 0, 0.5, 1, 2, 3 and 4. The $^{1}$H,$^{15}$N-TROSY spectra of $^{15}$N-labeled integrin from all the titration points were overlaid and plotted. The intensity changes for selected resonances, all thought to be at or in the vicinity of the Talin-F3 binding site, were plotted (bottom panels) versus concentration and the $K_d$ was obtained by a single global fit of all the data. left: integrin β1 TM/CT WT, right: integrin β1 K752E TM/CT.

reduces the affinity of the talin F3 domain for the isolated integrin subunit. The affinity of F3 for both the β1 and β3 TM/CT is higher by over an order of magnitude than the affinity of F3 for the isolated CT of these integrins in buffer minus either membranes or a membrane-mimetic ($K_d$ of 490 μM and 270–600 μM for β1 and β3, respectively) (*Moore et al., 2012*; *Saltel et al., 2009*; *Anthis et al., 2010*). On the other hand, it should be acknowledged that the full affinity of the complex between talin and the integrin β3 CT in membranes containing an anionic lipid (Kd ca. 0.9 μM) requires the presence of a linked talin F2 domain (*Moore et al., 2012*).

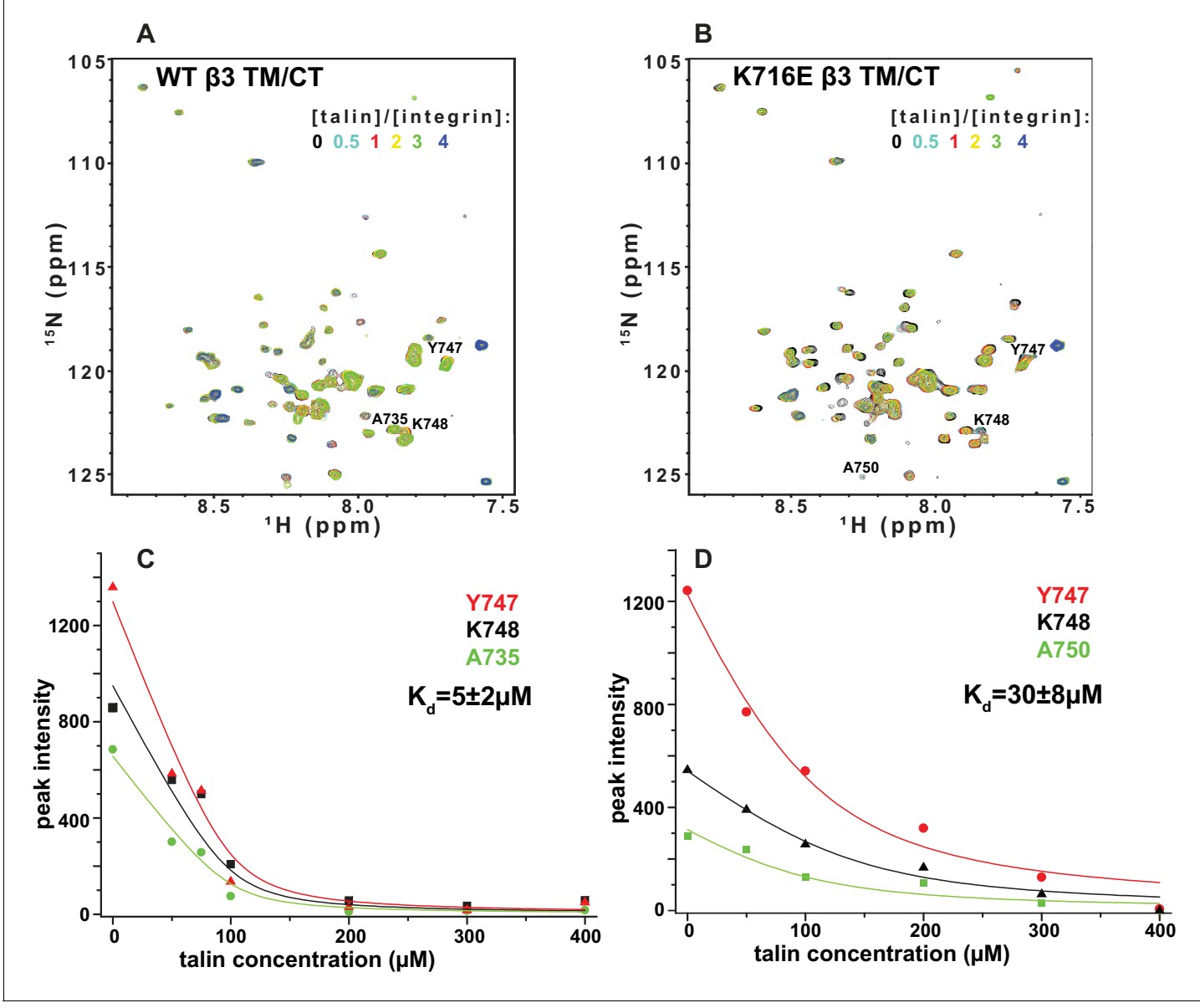

**Figure 10.** NMR-monitored titrations of integrin β3 TM/CT by talin. [15]N-labeled WT/K716E integrin β3 TM/CT was titrated with unlabeled talin1-F3. The concentration of all integrins was 100 μM. The samples contained 5% q = 0.3 D6PC/POPC/POPS (POPC:POPS = 2:1) bicelles in 25 mM HEPES buffer at pH 7.4. The TROSY NMR spectra were collected at 35°C at 900MHz (upper panels). In both cases six titration points were collected in which the molar ratio between talin1-F3 and the integrin was varied: 0, 0.5, 1, 2, 3 and 4. The [1]H,[15]N-TROSY spectra of [15]N-labeled integrin from all the titration points were overlaid and plotted. The intensity changes for selected residues, all thought to be at or in the vicinity of the Talin-F3 binding site, were plotted (bottom panels) against concentration and the $K_d$ was obtained by a global fit of all the data. Left: integrin β3 TM/CT WT. Right: integrin β3 K716E TM/CT.

## Discussion

The goal of this study was to determine whether the canonical mechanism of integrin activation that has emerged from studies of the platelet specific αIIbβ3 integrin extends to the ubiquitously expressed β1 integrins. We show that the structures of the β1 and β3 CT/TM in bicelles are distinct. Also, the highly conserved 'snorkeling lysine' that plays a key role in regulating αIIβ3 activation plays an opposite role in β1 integrin function and does not play a role in defining the tilt of either β1 or β3. Very different affinities exist between β1 and some α subunits relative to those observed

between αIIb and β3. We therefore conclude that the diverse biophysical properties of β1 and β3 integrin TM and CT domains in combination with varying α subunit properties result in distinct mechanisms of integrin activation and function.

In this study, we show that, in contrast to integrin αIIbβ3 expressed in a CHO cell, the β1-K752E mutation does not promote integrin activation (*Kim et al., 2012*) but actually decreases integrin α1β1, α2β1 and α5β1 epithelial cell adhesion to their respective ligands, with little effect on cell spreading. Consistent with this, β3 tail mutations that resulted in normal integrin activation and talin binding but decreased cell spreading, were recently described (*Pinon et al., 2014*). A previous study showed that expression of β1-K752L in integrin β1-null fibroblasts did not alter the integrin activation status or the ability of the cells to adhere to fibronectin and laminin-111; however this mutation did inhibit cell spreading and migration (*Armulik et al., 2004*). The mechanism for the altered integrin β1-K752L-mediated cell function was proposed to be due to altered phosphatidylinositol 3-kinase activation that impacted FAK-independent integrin signaling without altering integrin activation (*Armulik et al., 2004*). We did not test the effect of a β1-K752L mutation in CD cells. Nevertheless, a possible reason for the differences in integrin activation in cell systems that are null for β1 and those in CHO cells is that in the former, the mutant integrin is transduced into a β1-null CD cell, while in the CHO cell system the mutated β integrin is expressed into a cell that already has endogenous integrins (*Kim et al., 2012*). Alternatively, it could simply reflect differences in cell type. Another important point is that the only effect of the mutation of the 'snorkeling lysine' tested in CHO cells was on integrin activation, and not cell functions that rely on ligand-dependent integrin signaling such as adhesion, migration or spreading.

Another key observation is that the β1-K752 and β3-K716 mutations do not seem to play an important role in defining the topology of the TM in the free subunits. β3-K716 was previously believed to stabilize the integrin αIIbβ3 inactive state by forming a salt bridge between its amino side chain and anionic phosphodiester linkages in the phospholipid head groups (*Kim et al., 2012*). This interaction has been proposed to be a key determinant of the TM tilt that is required for optimal outer and inner clasp interactions between the αIIb and β3 subunits, which are present only in the inactive state (*Kim et al., 2012*). This led us to conduct Mn(II)-induced paramagnetic relaxation NMR experiments (PRE) on WT and β1-K752E to determine whether K752 also promotes tilt of the TM. To our surprise, we observed that β1-WT and β1-K752E integrin exhibit the same PRE patterns, indicating that the tilt of the β1 integrin TM is independent of the charge of residue 752. Paradoxically, we made the exact same observation for the β3-K716E mutation indicating that neither β1 nor β3 integrins form a lysine/phospholipid ion pair critical for promoting TM tilt. Our studies revealed that previously reported PRE results to the contrary had been subject to an experimental artifact, as detailed in the Results section. Hence, the previous work had mistakenly concluded that the different PRE patterns observed for β3-WT and β3-K716E reflected very different membrane topologies for WT and mutant (*Kim et al., 2012*). Here, we repeated the original experiment, but took pains to eliminate the artifactual interaction, revealing that both WT and the K-to-E mutant forms of both β3 and β1 have membrane topologies that are similar in all four cases. We confirmed these observations by reciprocal NMR PRE experiments using a lipophilic paramagnetic probe. Thus, not only does β1-K752 promote functional output that is opposite to that promoted by β3-K716, but even in the case of β3 the mechanism by which β3-K716 promotes the inactive signaling state must now again be regarded as unknown. Our results are consistent with another study that used a biotin maleimide chemical modification of engineered Cys side chains to show that the K716P mutation in β3 did not result in any change in membrane topology (*Kurtz et al., 2012*).

Given that the functional consequences of mutating β3-K716 and β1-K752 seem to arise from different mechanisms than originally thought, we tested whether the K-to-E mutations alter association of the integrin β tails with talin, an interaction that promotes integrin activation. Consistent with previous work (*Anthis et al., 2010*), we found that the talin F3 domain binds to the WT β3 TM/CT significantly more tightly than to WT β1 TM/CT (Kd for β3 lower by a factor of 5). Interestingly, the β3-K716E mutation increased Kd for talin binding by a factor of 6, a finding that that does not correlate well with the proposed activation-promoting nature of this mutation. On the other hand, the β1-K752E mutation alters Kd by less than 2-fold and does so in favor of tighter binding. While possible roles for the F0-F2 talin domains in possible coupling between the TM lysine site and talin binding were not tested in this study, at face value these results suggest that the impact of the TM K-to-E mutation on talin binding is not the basis for how this mutation alters integrin function, once again

suggesting a mechanism whereby the TM K-to-E mutation alters integrin function by altering integrin-dependent signaling and not activation. It is interesting that talin binds to both integrin TM/CT domains in bicelles more tightly than to CT-only domains in buffer (*Moore et al., 2012*; *Anthis et al., 2010*). Bicelles most likely promote a more native-like conformational state of the CT in the context of its attached and bicelle-anchored TM than for the isolated CT in solution.

This work also reveals significant differences in the secondary structures of the β1 and β3 TM/CT, which were here examined under identical bicellar conditions. The β1 TM ends after I758 while the β3 TM ends after I721, in reasonably good accord with previous NMR, EPR, and biochemical studies. The TM helix of both β1 and β3 was seen to extend into the cytosol and through the juxtamembrane residues that form the inner clasp, albeit with some fraying. However, unlike β1, where the helix ends with the clasp at K765, the helix in the β3 extends a full 10 residues further into the subunit, ending only at A737 (*Figure 2C*). This result is reminiscent of the observations that a continuous β3 TM/CT extends to D740 as part of the αIIbβ3 TM/CT heterodimer in an organic solvent mixture (*Yang et al., 2009*), while in the corresponding complex for the CT-only heterodimer the β3 helix ends at K738 (*Vinogradova et al., 2002*). A previous cross-linking/computational study of the αIIbβ3 complex in cell membranes also indicated the β3 helix extends from the TM far out into the cytosol (*Zhu et al., 2009*). Our results are also generally consistent with a previous NMR study of the β3 TM/CT domain in DPC micelles, where an extended CT helix was observed, but was connected to the TM by a flexible linker (*Li et al., 2002*). Our data also support the presence of hinge motion centered at the TM/CT interface, although our data suggest that hinge formation is transient, not a stable structural feature. Other NMR or EPR structural studies of the β1 or β3 TM in isolation or of the αIIβ3 TM complex did not include the full CT (*Moore et al., 2012*; *Lau et al., 2008a*; *Yu et al., 2015*). The combined structural and membrane topological studies of our work also offer clear evidence that, on the average, the CT helix for both β1 and β3 TM/CT extends away from the membrane surface into the aqueous phase. Our results are not consistent with the notion that the CT has significant membrane surface affinity, as suggested based on a previous NMR study of crosslinked αIIb and β3 CT in DPC micelles (*Metcalf et al., 2010*).

The studies of this work were conducted in *model* membranes and using excised TM/CT domains rather than full length integrins in cellular membranes. Both of these facts hinder the extrapolation of the observations to full length integrins in native membrane conditions. Nevertheless, they do suggest that some of the functional differences between β1 and β3 are linked to the different intrinsic conformational preferences of their CT, which likely impacts their selectivity and affinity in engaging their cytosolic effector proteins. It is interesting to note that while we observed the β3 helix to extend through site A737, in an NMR structure of the complex of the β3 CT with the talin F3 domain the helix terminates at amino acid 732 (*Wegener et al., 2007*), suggesting destabilization of the C terminal end of the helix by talin. Conversely, for bicelle-associated β1 the helix was seen to terminate at K765, while in a crystal structure of the β1 CT with the talin F2F3 domains this helix does not terminate till A773 (*Anthis et al., 2009*). These results suggest that the end of the β3 TM/CT helix is not very stable but is readily disrupted by events such as engagement by talin. This is consistent with the fraying of the CT helix seen in the results of this paper. At the same time the disordered segment C-terminal to the β1 TM/CT helix does have helical propensity that is manifested upon complex formation with talin. The metastability of secondary structure in both β1 and β3 CT seems well suited to enable optimal interactions to cytosolic binding partners.

Finally, the data demonstrated that the interactions of different α subunit TM/CT with the β1 TM/CT are characterized by very different affinities, ranging from very weak interactions between α1 or α2 and β1 to much higher affinity interaction between α5 and β1, similar to that found between αIIb and β3. Based mostly on studies of the αIIbβ3 integrin it has been widely assumed that the TM/CT of β integrins have an intrinsic affinity for the corresponding domains of their cognate α subunits, such that they will form constitutively inactive heterodimers. Many studies have shown the isolated αIIb and β3 TM associate to form heterodimers in model membranes or as fusion proteins in *E. coli* or model cell lines (*Lau et al., 2009*; *Berger et al., 2010*; *Partridge et al., 2005*; *Zhu et al., 2010*; *Schneider and Engelman, 2004*; *Schmidt et al., 2015*; *Lokappa et al., 2014*; *Kim et al., 2009*). We observed similar results for heterodimerization of the α5 and β1 TM/CT, an observation consistent with evidence that this particular β1 integrin is activated according to the canonical model (*Takagi et al., 2003*). In contrast, we found that α1 and β1 as well as α2 and β1 TM/CT interactions were too weak to be quantified in bicelles, even at the high protein concentrations required for

NMR spectroscopy. This is surprising in light of studies suggesting that the fusion proteins containing the TM-only domain of these integrin subunits can form heterodimers in *E. coli* (*Berger et al., 2010*; *Schneider and Engelman, 2004*). However, these latter studies were conducted in the absence of the β1, α1, and α2 CT, which almost certainly profoundly impact heterodimerization (*Briesewitz et al., 1995*; *Liu et al., 2015*). Our results suggest that the α1 and α2 CT may actually inhibit formation of α1β1 and α2β1 TM/CT heterodimers, at least in bicelles. The stark contrast between the collagen α1β1 and α2β1 integrins and the fibronectin α5β1 integrin suggests that the role of TM/CT domain heterodimerization in regulating integrin function may vary considerably among different β1 integrins, as previously proposed (*Nissinen et al., 2012*; *Abair et al., 2008b*; *Bazzoni et al., 1998*; *Pepinsky et al., 2002*; *Bodeau et al., 2001*).

Our results for integrins α1β1 and α2β1 suggest the intriguing possibility that these receptors may remain constitutively in their unclasped α/β-TM/CT-dissociated forms, implying their signaling functions are modulated via mechanisms other than the canonical model of switching between TM/CT-clasped and unclasped forms (*Abair et al., 2008a*; *Nissinen et al., 2012*). That some integrins may be constitutively active or unclasped has long been postulated (*Bazzoni and Hemler, 1998*). Determining whether this is actually the case will require additional studies. However, even considering that the energetics of heterodimerization of isolated TM/CT will not be the same as local TM/CT heterodimerization in the context of full length integrin subunits, these results indicate that the energetics of TM/CT heterodimerization vary dramatically from integrin to integrin.

In conclusion, we present evidence that while integrin αIIbβ3 is found in both active and inactive conformations, a subclass of β1 integrins (α1β1 and α2β1) may adopt a constitutively active conformation. Thus β1 and β3 integrins appear to have distinct mechanisms of action wherein different modes of integrin regulation likely occur within the β1 integrin class based on which α subunit is involved, as well as which cell type. Such cell type-specific mechanisms of integrin function need to be explored if we are to understand how different integrins function in distinct biological settings.

# Materials and methods

## Cell adhesion

Cell adhesion assays were performed in 96-well plates as previously described (*Chen et al., 2004*). Cells ($1 \times 10^5$) were seeded in serum-free medium onto plates containing different concentrations of ECM for 60 min. Adherent cells were fixed, stained with crystal violet, and solubilized, and the optical densities of the cell lysates were read at 570 nm (OD570). Human placental collagen IV and rat tail collagen I were purchased from Sigma-Aldrich, St Louis, MO.

## Cell Spreading

Coverslips were coated with either collagen I (0.5 μg/ml), collagen IV (0.25 μg/ml) and fibronectin (0.5 μg/ml) and blocked with 2% heat inactivated BSA. Cells were seeded in serum-free medium and incubated for either 15 min, 30 min or 45 min after which they were fixed with 4% paraformaldehyde and stained with Rhodamine labeled Phalloidin. Images were collected with confocal microscopy and the cell area was determined using ImageJ software.

## Integrin activation assays

The active conformation of integrin β1 on adherent cells was determined using the 12 G10 antibody (Millipore MAB2247, Darmstadt, Germany) that specifically binds to the active conformation of integrin β1. Total surface expression was determined using AIIB2 antibody. CD cells stably expressing either WT or K752E integrin β1 were allowed to adhere for 1 hr to eight well chamber glass slides (Millicell EZ slides, Millipore, USA Cat no. PEZGS0816) coated with collagen I (20 μg/ml) or fibronectin (10 μg/ml) at 4°C overnight. Adherent cells were fixed with 10% formaldehyde, incubated with primary antibody (1:100) followed by secondary antibodies (1:100) and visualized using a Zeiss LSM 510 microscope. Images were taken close to the substrate. The intensities of images were analyzed using ImageJ software (JACoP). Manders' overlap coefficient based on the Pearson's correlation coefficient for average intensity values was in each case quantified and expressed as a percentage of 12 G10 relative to AIIB2. Statistical plots show the mean values and SD of 30 cells per group.

## Plasmid construction

cDNA for the wild type human integrin β1 TM/CT (resides 719–798), integrin β3 TM/CT (residues 685–762), integrin β3 transmembrane-only domain (TM-only) (residues 685–727) and talin1-F3 domain (residues 309–405) were sub-cloned into a pET16b vector (Novagen, Darmstadt, Germany), which adds an N-terminal His$_6$ purification tag (MGHHHHHHGM-). The single native cysteine present in each construct—C723 in integrin β1 TM/CT, C687 in integrin β3 TM/CT, and C336 in the talin1-F3 domain—were mutated to serine in order to avoid aberrant disulfide formation. No adverse effects of the C687S mutation in β3 or of the C336S mutation in talin1 were reported in previous studies of these proteins (*Lau et al., 2008a*, *2008b*; *Song et al., 2012*) (*Ulmer et al., 2003*). QuikChange site-directed mutagenesis (Stratagene, La Jolla, California) was used to produce each of the following variants: K752E integrin β1 TM/CT, K716E integrin β3 TM/CT, and the K716E integrin β3 TM-only.

## $^{15}$N-amino acid-specific labeling

Amino acid-specific isotopic labeling is obtained by using an auxotroph *E. coli* strain, CT19, which is incapable of synthesizing Leu/Ile/Val/Ala/Tyr/Phe/Trp/Asp. The cells depend solely on diet to get these amino acids, enabling supplementation of the medium with one of these amino acids in $^{15}$N-labeled form, leading to specific labeling. The plasmid encoding the integrin construct was transformed into CT16 cells and plated on LB agar with three antibiotics used for selection: 100 mg/L ampicillin, 100 mg/L kanamycin, and 20 mg/L tetracycline. The plate was incubated overnight at 37°C. As tetracycline is light sensitive, the plate and all the media flasks were covered with aluminum foil. A single colony was used to inoculate 5 ml LB media with those three antibiotics. After overnight growth at 37°C, 1 ml of LB cell culture was used to inoculate 1 L M9 media at room temperature with the addition of 0.5 g of each of the eight acids deficient in the CT16 auxotroph. After OD600 reached 0.8 the cells were spun down at 3000 rpm for 15 min and transferred to the final medium at room temperature which contained 0.2g of the desired $^{15}$N-labeled amino acid and 0.5 g unlabeled each for other 7. The cells were then induced by adding IPTG to 1 mM and harvested after overnight growth. The protein was purified following the protocol in the Materials and methods section.

## $^{15}$N-Amino acid reverse labeling

This method results in one amino acid type remaining unlabeled in a protein under conditions in which all other amino acids are $^{15}$N-labeled. This will lead to $^{1}$H,$^{15}$N-TROSY or HSQC spectra in which all peaks are seen except for the amino acid that was deliberately not $^{15}$N-labeled. Our experience shows this labeling strategy works well for Arg, Lys, and His. The labeling strategy is similar to standard uniform $^{15}$N-labeling in $^{15}$N-enriched minimal medium with two differences: first, 1 g of the unlabeled amino acid is added to the liquid culture at room temperature immediately before induction and second, cells are harvested 8 hr after induction and continued incubation at room temperature. The protein was purified using the protocol given below.

## Purification of integrin β1 and β3 TM-only and TM/CT domains

BL21(DE3) cells were transformed with the pET16b vectors. M9 cultures were grown at room temperature to an OD$_{600}$ of 1.0 prior to induction with 1 mM IPTG. After 24 hr of induction at room temperature, the cells were harvested by centrifugation and stored at −80°C for future use. $^{15}$N-NH$_4$Cl, $^{13}$C-glucose, and/or D$_2$O were incorporated into the M9 media in order to produce proteins with the desired isotopic labeling for NMR spectroscopy.

For protein purification frozen packed *E. coli* cells were suspended in lysis buffer (75 mM Tris-HCl, 300 mM NaCl and 0.2 mM EDTA, pH 7.7) at 20 ml per gram of cells. Next 5 mM Mg(Ac)$_2$, 0.2 mg/ml PMSF, 0.02 mg/ml DNase, 0.02 mg/ml RNase and 0.2 mg/ml lysozyme were added to the cell slurry, which was tumbled at room temperature for 1 hr. The lysate was then sonicated for 10 min at 4°C. Following sonication, the detergent Empigen (Sigma-Aldrich, St Louis, MO) was added to 3% (w/v) and the mixture was again tumbled at 4°C for 1 hr to solubilize the integrins. Insoluble debris was then removed by centrifugation at 20,000 g for 20 min. The supernatant was collected and incubated with Ni$^{2+}$-NTA resin (Qiagen, Valencia, CA) (1 ml per liter of cell culture) at 4°C for 1 hr. The resin was collected by centrifugation at 3700 g for 5 min and packed into a chromatography column, which was washed with buffer A (40 mM HEPES, 300 mM CHPNaCl, pH 7.5) containing 3%

Empigen followed by wash buffer (buffer A plus 40 mM imidazole (IMD) and 1.5% Empigen) until the $A_{280}$ returned to the baseline level. Empigen on the column was then exchanged with the mild detergent dihexanoylphosphatidylcholine (D6PC, Avanti Polar Lipids, Alabaster, Al, USA) by washing the column with 8 × 1 column volumes of exchange buffer (100 mM NaCl 10 mM IMD, pH 7.4) containing 2% D6PC, and further exchanged into bicelles by washing the column with two column volumes of exchange buffer containing 2% w/v bicelles and 10% $D_2O$. Integrins were then eluted in 250 mM imidazole (pH 7.4) containing 2% bicelles and 10% $D_2O$ (w/w). Two different bicelle compositions were used: D6PC/DMPC q = 0.3 and POPC/POPS/D6PC (POPC:POPS = 2:1) q = 0.3, where q is the lipid to detergent mole ratio. The eluted protein solution was concentrated 10-fold by using an Amicon Ultra centrifugal filter cartridge with a molecular weight cut-off of 10 kDa and the pH was adjusted to 6.5 using acetic acid. 200 µl was then transferred to a 3 mm NMR tube. NMR samples prepared in this manner contained 10% $D_2O$ for field frequency locking purposes, 1 mM EDTA, 250 mM IMD, pH 6.5. The final bicelle concentration was 20% (w/w) and the integrin concentration was typically 0.5 mM.

For some samples, IMD was exchanged out as buffer for HEPES. To accomplish this, the concentrated protein solution above (with 250 mM IMD) was diluted 10-fold in 25 mM HEPES pH 7.4 containing 0.7% D6PC (equal to its critical micelle concentration, CMC) and 10% $D_2O$, then concentrated again 10-fold. Two additional rounds of dilute/centrifugation buffer exchange were carried out to reduce the final imidazole concentration to <1 mM. The 0.7% D6PC present in the exchange buffer was employed to maintain a constant q by offsetting loss of the free population of D6PC (equal to its CMC) through the centrifugal filter. Finally, EDTA was added from a 200 mM stock in 25 mM HEPES buffer at pH 7.4 to a concentration of 1 mM in the NMR sample. By this method the final NMR samples contained 10% $D_2O$, 1 mM EDTA, 25 mM HEPES, pH 7.4. The final bicelle concentration was 20% (w/w) and the integrin concentration was typically 0.5 mM, which is 0.57% molar percent (mol%) relative to total moles of DMPC.

Talin1-F3 was purified in a similar manner except that no lipids or detergents were added to any of the purification buffers. Briefly, cells were lysed in an identical manner as described above prior to centrifugation at 20,000 g for 20 min in order to remove cellular debris. The lysate was then incubated with $Ni^{2+}$-NTA resin (Qiagen, Chatsworth, CA) (1 mL per liter of cell culture) at 4°C for 1 hr. The resin was then collected by centrifugation at 3700 g for 5 min and packed into a chromatography column. The column was first washed with buffer A followed by buffer A containing 40 mM imidazole. After $A_{280}$ returned to baseline, the talin1-F3 was eluted in 500 mM IMD (pH 7.4). The talin1-F3 was concentrated to 0.8 mM by using an Amicon Ultra centrifugal filter cartridge with molecular weight cut-off of 3 kDa and stored at 4°C for future use. Prior to NMR studies buffer exchange was carried out by 3 rounds of 10-fold dilution/centrifugal concentration with an exchange buffer (25 mM HEPES in 10% $D_2O$, 1 mM EDTA, pH 7.4). The final talin1-F3 stock concentration was 0.8 mM.

Protein concentrations were determined from $A_{280}$ using extinction coefficients calculated by the 'protparam' program (http://web.expasy.org/protparam/). The pH was adjusted using either acetic acid or ammonium hydroxide.

## Expression and purification of $\alpha$ integrin TM/CT

The combined TM/CT of integrin α1 (residues 1132–1179), α2 (1125 to 1181), αIIb (9189 to 1039), and α5 (990 to 1050) were expressed and purified as described previously (*Ulmer et al., 2003*; *Lu et al., 2012*; *Mathew et al., 2012b*). These recombinant proteins were eluted from the Ni resin in the presence of 250 mM imidazole and bicelles and prepared for biophysical studies as described above for the integrin beta TM/CT.

## Backbone NMR resonance assignments

NMR data were collected for the β1 and β3 TM/CT subunits at 35°C or 45°C on Bruker 900, 800 and 600 MHz spectrometers equipped with cryogenic triple-resonance probes with z-axis pulsed field gradients. $^1H$, $^{15}N$-TROSY experiments were conducted using the standard Bruker pulse sequence trosyetf3gpsi (version 12/01/11), which makes use of a sensitivity enhanced, phase-sensitive TROSY pulse sequence and WATERGATE solvent suppression. The integrin β1 TM/CT NMR samples contained 0.5 mM (0.57 mol% relative to moles of DMPC) uniformly $^{13}C$, $^{15}N$, $^2H$-labeled integrin β1 TM/CT, 20% D6PC/DMPC q = 0.3, 1 mM EDTA, 250 mM IMD at pH 6.5% and 10% $D_2O$. The

integrin β3 TM/CT NMR samples had the same composition as the β1 sample except that the protein was not perdeuterated. The following TROSY-based 3D experiments were collected in order to assign backbone resonances: HNCA, HN(CO)CA, HNCACB, HNCO, HN(CA)CO, NOESY-TROSY-HSQC. Selective $^{15}$N-labeling of specific amino acids was also employed to facilitate resonance assignments. NMR data were processed using NMRPipe (*Delaglio et al., 1995*) and analyzed using NMRViewJ (*Johnson, 2004*). Backbone chemical shift assignments were analyzed using chemical shift index analysis (*Wishart and Sykes, 1994*) and TALOS-N (*Shen and Bax, 2013*) in order to determine the secondary structure. The TROSY peak assignments of the mutant protein or WT protein under pH 7.4 D6PC/POPC/POPS bicelles conditions were obtained by comparing spectra to the pH 6.5 'reference' assigned WT spectrum. For some mutants/conditions a 3D TROSY-HNCA was collected and analyzed to assist with confirming and completing assignments.

## NMR backbone amide/water hydrogen exchange and $^{15}$N transverse relaxation rate (R2) measurements

The CLEANEX-PM NMR experiment (*Hwang et al., 1998*) was carried out using a mixing time of 100 msec to map the extent of hydrogen exchange between backbone amide sites and water occurring over a 100 msec time scale. $^{15}$N relaxation experiments on $^{15}$N-labeled integrin TM/CT were carried out at 45°C on Bruker Avance NMR spectrometers. A TROSY-based version of the Carr-Purcell-Meiboom-Gill experiment was used to obtain $^{15}$N transverse relaxation times (T2), which are presented as relaxation rates (R2=T2$^{-1}$) (*Zhu et al., 2000*). The transverse magnetization decay was sampled at nine points: (17, 35, 52, 69, 86, 104, 138, and 173 msec). The data were recorded in pseudo-3D mode with a recycle delay of 3 s. All relaxation spectra were recorded with 1024 × 128 complex data points and the spectra widths were 13 by 23 ppm in the $^{1}$H and $^{15}$N dimensions, respectively.

## Probing the membrane topology of β1 and β3 subunits

The membrane topologies of the following six proteins were probed: integrin β1 TM/CT, integrin β1 K752E TM/CT, integrin β3 TM/CT, integrin β3 K716E TM/CT, integrin β3 TM-only, and integrin β3 K716E TM-only. For each, two sets of sample conditions were used: one using fully zwitterionic bicelles containing 20% D6PC/DMPC bicelles q = 0.3, with 1 mM EDTA 250 mM IMD at pH 6.5 in 10% D$_2$O. The other conditions employ net-negatively charged bicelles composed of 20% D6PC/POPC/POPS (POPC:POPS = 2:1), q = 0.3, with 25 mM HEPES at pH 7.4 in 10% D$_2$O. The protein concentrations were ~0.5 mM (0.57 mol%) for integrin β1 and ~0.3 mM (0.34 mol%) for integrin β3. Two paramagnetic probes were employed; the lipophilic 16-doxyl stearic acid (16-DSA) probe (Santa Cruz, San Diego, CA, USA) and the water-soluble Gd-DTPA probe (Santa Cruz, San Diego, CA, USA). For the sake of comparison with Gd-DTPA, in the cases of the integrin β3 TM-only and the β3 K716E TM-only, Mn-EDDA was utilized as a water-soluble probe following the published protocol (*Lau et al., 2008a*).

The 900 MHz $^{1}$H, $^{15}$N-TROSY spectrum of each $^{15}$N-integrin was monitored to quantitate the line-broadening effects of different paramagnetic probes. For the water-soluble paramagnetic probes, a stock solution (50 mM for Mn-EDDA and 500 mM Gd-DTPA in the same buffer and pH as NMR samples) was made and directly added to NMR samples (to 1 mM for Mn-EDDA or 10 mM for Gd-DTPA). 16-DSA was first dissolved in methanol and the appropriate amount was transferred to an Eppendorf tube and dried under vacuum for 4 hr prior to solubilizing in integrin/bicelle NMR solutions. The final molar concentration of 16-DSA in bicelles was 2.5 mM (4 mol% of total lipid).

Matched TROSY spectra of $^{15}$N-labeled integrins were collected in the presence and absence of each paramagnetic probe. The intensity ratios for the corresponding peaks in the pairs of spectra were calculated as an indicator of the probe access to the amide site. The spectra were processed using NMRPipe (*Delaglio et al., 1995*) and analyzed with NMRViewJ (*Johnson, 2004*)

## Quantitation of heterodimerization affinity between α and β integrin TM/CT by NMR spectroscopy

The formation of heterodimers between different integrin β and α subunits during subunit titration was monitored by $^{1}$H,$^{15}$N-TROSY spectroscopy at 45°C. The following integrin TM/CT complexes were probed in this study: (i) the heterodimers formed by either WT or K752E mutant forms of β1

with WT forms of α1, α2 and α5, (ii) the heterodimers formed by either WT or K716E forms of β3 with wild type αIIb. All titration points maintained the same model membrane and buffer conditions: 20% (w/v) D6PC/POPC/POPS (POPC:POPS = 2:1), q = 0.3, 50 mM phosphate buffer with 1 mM EDTA in 10% $D_2O$, pH 6.5. All the purified proteins were concentrated by a factor of 10 using centrifugal concentrators (Amicon Ultra, 10kDal cut-off) to attain 20% bicelles in the samples. Protein concentration in mol% was calculated as the (moles of protein X 100)/(total moles lipid), where D6PC (a detergent) is not considered to be a lipid.

Integrin TM/CT complex formation for αIIbβ3(WT), αIIbβ3(K716E), α5β1(WT) and α5β1(K752E) were monitored by acquiring $^{1}H$-$^{15}N$-TROSY spectra on a Bruker 900 MHz NMR spectrometer. Rather than titrating a single β subunit sample with multiple aliquots from a stock solution of the alpha subunit, six separate NMR samples were prepared for each titration at mole ratios of α/β = 0, 0.5, 1, 2, 3, and 4. The concentrations of WT β3, WT β1 and the K752E β1 mutant were fixed at 0.17 mol% (150 μM) for all the samples, while the concentration for the K716E β3 mutant was fixed at 0.10% (90 μM). For these titrations, no peak shifts were observed, but some β1 $^{1}H$-$^{15}N$-TROSY peaks disappeared during the course of the titrations, indicating subunit association/dissociation exchange that is slow on the NMR time scale. Peak intensities were plotted against the concentration of the unlabeled α subunit for each titration. Variations in peak intensities as a function of the α subunit were globally fit by a 1:1 binding model similar to that given in the next paragraph using Origin-Pro9.0 software.

Determination of the dissociation constants for integrin α1β1(WT), α2β1(WT), α1β1(K752E) and α2K752Eβ1(K752E) TM/CT complexes was based on data from titrations monitored using a Bruker 600MHz NMR spectrometer. Six NMR samples were prepared for each titration, at varying molar ratios of α/β: 0, 1.25, 2.5, 3.75, 5, and 7.5. The concentrations of the WT and K752E mutant integrin β1 subunits were fixed at 0.23 mol% (200 μM) for all samples. For these titrations on/off binding exchange was rapid on the NMR time scale, such that α1 and α2 subunit-induced chemical shift changes in the spectra were monitored. Hybrid $^{1}H$ and $^{15}N$ amide shifts at each point were measured using the following equation (*Ayed et al., 2001*):

$$\Delta_{(HN)} = \sqrt{\Delta_H^2 W_{H+}^2 \Delta_N^2 W_N^2} \tag{1}$$

Where $W_H = 1$ and $W_N = 0.154$ are weighting factors for the $^{1}H$ and $^{15}N$ amide shifts, respectively.

The hybrid chemical shift changes were plotted against the concentration of unlabeled integrin α1 TM/CT or integrin α2 TM/CT and fit by the 1:1 binding model using OriginPro9.0 nonlinear regression:

$$\Delta_{obs} = \Delta_{max} \frac{(K_d + [L_0] + [U_0]) - \sqrt{(K_d + [L_0] + [U_0])^2 - (4[U_0][L_0])}}{2[U]_0} \tag{2}$$

where $K_d$ is the dissociation constant, $\Delta_{obs}$ is the observed hybrid chemical shift change, $\Delta_{max}$ is the shift difference between free subunit and saturated complex conditions, and $[U_0]$ and $[L_0]$ are the mol% concentrations of unlabeled and labeled proteins, respectively (*Anthis et al., 2010*).

## Quantitation of heterodimerization affinity between α and β integrin TM/CT by steady state fluorescence anisotropy

The affinities by which integrin β1 and β3 wild type and K752E/K716E mutants form heterodimer complexes with α subunits were determined from steady state fluorescence anisotropy measurements. Single cysteine mutants of the α subunit TM/CT (α1 L1142C, α2 T1132C, α5 E992C and αIIb E960C) were labeled with the fluorophore IAEDAN and titrated with unlabeled cysteine-free β1 and β3 wild type and K752E/K716E mutants in 2% q = 0.3 D6PC/POPC/POPS (POPC/POPS = 2:1) bicelles in 25 mM HEPES buffer pH 7.4 at 35°C.

Protein preparation and labeling for the above measurements was carried out as follows. His$_{6}$-tagged mutant single-cysteine α integrin TM/CT were originally eluted from the Ni-NTA column into 0.5% DPC micelles in 250 mM imidazole, pH 7.8. The protein concentration was assessed by measuring $A_{280}$ and then adjusted to ca. 50 μM by diluting with the elution buffer (0.5% DPC micelles, pH 7.8 and 250 mM imidazole), protein being quantitated via $A_{280}$ measurement. EDTA was added

from a stock solution to 1 mM and the pH was adjusted to 6.5 with acetic acid. DTT was then added to a concentration of 2.5 mM and the solution was placed under argon. The solution allowed to mix at room temperature (RT) with gentle shaking for 16–20 hr. DTT and imidazole were removed using a PD-10 desalting column pre-equilibrated with 20 mM Tris, 150 mM NaCl, 0.5% DPC, pH 7.8. A 10X molar excess of IAEDAN (from a stock solution 25 mM acetonitrile) was then added to the protein solution based on the assumption that the protein concentration was ca. 50 μM. The reaction of IAEDAN with the protein TM/CT thiol group was allowed to proceed at room temperature under argon for 3 hr. Excess free label was then removed by desalting over a PD-10 column in 20 mM Tris, 150 mM NaCl, 0.5% DPC pH 7.8. Protein concentration and labeling efficiency were calculated using $A_{280}$ and $A_{373}$ (with $\varepsilon_{337} = 5600$ $M^{-1}cm^{-1}$ for IAEDAN). Labeled proteins were then re-bound to Ni NTA resin via incubation for 1 hr at RT. The labeled proteins on the resin was then washed with 16 column volumes of 0.5% DPC micelles in HEPES buffer 25 mM, pH 7.2 to further remove unreacted label. The resin-bound integrin was then re-equilibrated with 1% bicelles in HEPES buffer 25 mM, pH 7.2 using four column volumes of buffer and then eluted with 250 mM imidazole, 2% bicelles pH 7.8. EDTA was added to 1 mM and the buffer was exchanged via centrifugal filtration with 25 mM HEPES buffer, pH 7.4 containing 2% [D6PC/POPC/POPS (2:1)] bicelles. The protein concentration was determined using $A_{280}$. For use in the titrations unlabeled cysteine-free integrin β1 and β3 wild type and K752E/K716E mutants were eluted from Ni(II)-NTA resin using 250 mM imidazole, 2% D6PC/POPC/POPS bicelles pH 7.8. The eluted proteins were then buffer exchanged into 25 mM HEPES buffer, pH 7.4 containing 2% (D6PC/POPC/POPS) bicelles. The protein concentration was determined by $A_{280}$ before proceeding with anisotropy experiments.

0.2 μM IAEDAN-labeled single cysteine mutants of the α subunit TM/CT (α1 L1142C, α2 T1132C, α5 E992C and αIIb E960C) were titrated with unlabeled cysteine-free β1 and β3 wild type and K752E/K716E mutants in 2% D6PC/POPC/POPS bicelles in 25 mM HEPES buffer pH 7.4 at 35°C. Each titration point mixture was incubated for 16–20 hr at room temperature before conducting the anisotropy measurements. The mixture was allowed to equilibrate in the cuvette for 5 min before the start of experiments. Anisotropy measurements were carried out using a Horiba Jobin Yvon Fluoromax-3 fluorimeter equipped with an L-format, single cuvette holder with polarizers. Steady state anisotropy was measured using excitation and emission wavelengths of 337 and 487 at 35°C. Increases in anisotropy were measured as a function of the increasing concentration of unlabeled β TM/CT subunit to the IAEDAN-labeled α subunit. Origin 9.0 was used to fit a 1:1 stoichiometry binding model to the observed experimental anisotropy values versus the total concentration of unlabeled integrin β1 or β3, leading to determination of Kd. Concentrations were expressed in mol% (total moles of protein X100 / total moles of lipid in bicelles) units, as is appropriate for molecular association involving membrane-associated molecules. The detergent D6PC was not regarded as a lipid in these calculations.

## NMR titration of $^{15}$N-labeled integrin β1 TM/CT with unlabeled talin-F3

Four sets of titrations were carried out involving integrins (β1 TM/CT, β1 K752E TM/CT, β3 TM/CT, β3 K716E TM/CT) and the talin1-F3 domain. In all cases, the integrin was $^{15}$N-labeled and the concentration was fixed at 100 μM throughout the titration (using a series of individually prepared samples). The NMR spectra were measured in the presence of 0, 50, 100, 200, 300, 400 μM of unlabeled talin1-F3. The total detergent+lipid content was fixed at 5% (w/v) at a q-value of 0.3 (POPC/POPS/D6PC). The final solutions contained 25 mM HEPES pH 7.4% and 10% $D_2O$.

## Statistics

The Student's t-test was used for comparisons between two groups, and analysis of variance using Sigma Stat software was used for statistical differences between multiple groups. p<0.05 was considered statistically significant.

## Acknowledgements

The authors declare no competing or financial interests. The authors would like to thank Cathy Alford for performing the cell sorting. This research was funded by an AHA Scientist Development grant 16SDG29740001 to SM, VA Merit Reviews grant numbers 1I01BX002025 to AP, 1I01BX002196 to RZ; by the National Institutes of Health grant numbers R01-DK083187 to RZ and CS, R01-

DK075594 to RZ, R01-DK069221 to RZ, R01-DK095761 to AP Experiments using confocal microscopy were performed through the use of the VUMC Cell Imaging Shared Resource (supported by CA68485, DK20593, DK58404, DK59637 and EY08126). The NMR instrumentation used in this work was supported by NIH S10 RR026677 and NSF DBI-0922862.

## Additional information

### Competing interests

AP: Reviewing editor, *eLife*. RF: Reviewing editor, *eLife*. The other authors declare that no competing interests exist.

### Funding

| Funder | Grant reference number | Author |
|---|---|---|
| American Heart Association | Scientist Development 16SDG29740001 | Sijo Mathew |
| U.S. Department of Veterans Affairs | Merit Reviews 1I01BX002025 | Ambra Pozzi |
| National Institutes of Health | R01-DK095761 | Ambra Pozzi |
| National Institutes of Health | R01-DK083187 | Charles R Sanders Roy Zent |
| National Institutes of Health | S10 RR026677 | Charles R Sanders Roy Zent |
| National Science Foundation | DBI-0922862 | Charles R Sanders Roy Zent |
| U.S. Department of Veterans Affairs | Merit Reviews 1I01BX002196 | Roy Zent |
| National Institutes of Health | R01-DK083187 | Roy Zent |
| National Institutes of Health | R01-DK075594 | Roy Zent |
| National Institutes of Health | R01-DK069221 | Roy Zent |

The funders had no role in study design, data collection and interpretation, or the decision to submit the work for publication.

### Author contributions

ZL, SM, CRS, RZ, Conception and design, Acquisition of data, Analysis and interpretation of data, Drafting or revising the article; JC, Conception and design, Acquisition of data, Analysis and interpretation of data; AH, Conception and design, Acquisition of data; RP, Acquisition of data, Analysis and interpretation of data; BGH, RF, AP, Conception and design, Analysis and interpretation of data, Drafting or revising the article

### Author ORCIDs

Charles R Sanders, http://orcid.org/0000-0003-2046-2862
Roy Zent, http://orcid.org/0000-0003-2983-8133

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
