## [Decision Letter]

Thank you for submitting your article "Implications of the differing roles of the β1 and β3 transmembrane and cytoplasmic domains for integrin function" for consideration by *eLife*. Your article has been favorably evaluated by Fiona Watt (Senior Editor) and three reviewers, one of whom is a member of our Board of Reviewing Editors. The following individual involved in review of your submission has agreed to reveal his identity: Bernhard Wehrle-Haller (Reviewer #2).

All reviewers agreed that your study provides insight into an important and long-standing question in the field. However, there were a few points raised that, following discussion, we felt required addressing either experimentally or through more discussion of the data and approaches you use to recognise some potential caveats or alternative explanations. The Reviewing Editor has drafted this decision to help you prepare a revised submission, and details of the points we would like you to address are below.

Essential revisions:

1) The images of 12G10 staining alone are not sufficient to conclude anything at present – some form of formal quantification of intensity across multiple cells and experiments would be important to provide to support the conclusions from this data.

2) Analysis of spread cell area in the K752E mutant expressing cells across the different ligands would be important to show to determine whether the current results tally with those previously seen in CHO cells.

3) The lysine residue should be marked on the structural models in Figure 2, also the position of the membrane would help guide the reader better.

4) The dynamics data in Figure 2 again does not add a lot, without direct comparison to B3 it does not provide new insight and could be deleted.

5) Discussion, end of third paragraph: Yang et al. (Yang et al., 2009) shows R995 interacting with D723, not K716, so this statement saying that the data in that publication needs revisiting is not correct.

6) Please remove mention of 'structure' in the second paragraph of the subsection “The β1 and β3 TMD/CTD have different structures” as this refers to a structural model only.

7) The secondary structure outside the bicelle is surprising as helices do not usually form spontaneously without additional interactions; one interpretation might be that the helix is actually be binding back on to lipids. Please include some distance restraints, or residual dipolar coupling angle restraints to actually demonstrate that these secondary structure elements form in the absence of binding partners (or whether they are low propensity events as the fraying suggests). It seems unlikely that a single stranded β sheet would ever exist as is currently suggested; please include further discussion around this point.

8) The Anisotropy scale on Figure 6 is either incorrect (the change in anisotropy is 0.005 which is an incredibly small amount) or is not actually measuring a change in binding. Figure 6 is the only panel that gives a reasonable change in anisotropy (0.6) even though it shows the binding is weak. This is particularly important, as this is the only data that supports a major finding of this paper (constitutive activation of a1B1 and a2B1) and therefore the findings and potential caveats requires further discussion. Please also discuss why the Bmax of the anisotropy for the KE mutant higher: if it binds less well and has not reached saturation then one would expect less anisotropy?

9) The fluorescence anisotropy of the α1 and α2 might be due to an issue with the α tail peptide accessing the vesicle, or entering incorrectly. Just because the binding is weak with isolated peptides they may well interact (albeit weaker) in the context of the whole integrin with the result being that it might be more easily activated, however, that is not the same as being constitutively active. The authors should include discussion of this point in the revised study.

10) Given the concerns over the anisotropy data being insufficient to support the notion of differential activation, we request that additional chemical shift experiments are performed using β3+αIIb and β1+α1/α2/α5 tails to permit more transparent and direct comparison across the different heterodimers and help support the key conclusions relating to activation status of each integrin.

11) The probe accessibility measurements show no change with mutation which is surprising if the lysine is buried in the membrane. This data also does not give any readout on membrane tilt; no effect is seen but it is not clear if this is due to the experiment or the mutant. Without the α tail present it is not possible to judge the overall effect of this mutation as the lysine is positioned with the potential to make a number of contacts to phenylalanines on the α tail. It is possible that the lysine makes a cation-pi bond to the delocalized electrons of the phenylalanine ring and thus in a more physiologically relevant context the mutant might be more significant. The authors should discuss this point.

---

## [Author Response]

*Essential revisions:*

*1) The images of 12G10 staining alone are not sufficient to conclude anything at present – some form of formal quantification of intensity across multiple cells and experiments would be important to provide to support the conclusions from this data.*

We have quantified the 12G10 staining and it is present in the graphs in Figure 1 and in Figure 8. This was present in the last submission as well, however we mislabeled the graphs. This has been rectified in this submission.

*2) Analysis of spread cell area in the K752E mutant expressing cells across the different ligands would be important to show to determine whether the current results tally with those previously seen in CHO cells.*

We performed numerous spreading assays of the wildtype and mutant cells on different matrices. We show that the K752E mutant causes a moderate but significant decrease in spreading of collecting duct cells. These data are incorporated into the text of the revised manuscript and are shown in the new Figure 1 and Figure 8.

*3) The lysine residue should be marked on the structural models in Figure 2, also the position of the membrane would help guide the reader better.*

The positions of K752 and K716 are now indicated in Figure 2 for the B1 and B3 integrins, respectively. The experimentally determined positions of the transmembrane spans for these two proteins are indicated by blue color coding in this figure panel. However, we are reluctant to show these two proteins sitting in a membrane, as this would require us to depict the transmembrane segments as being either titled or un-tilted with respect to the bilayer normal. We hesitate to do this because the degree of tilt (or lack thereof) is not confidently determined by the data of our studies or, we would argue, by previous studies. We prefer to present these structural models conservatively rather than risk presenting a model that is not backed up by data.

*4) The dynamics data in Figure 2 again does not add a lot, without direct comparison to B3 it does not provide new insight and could be deleted.*

We have now collected transverse relaxation rates (R2) for the B3 integrin subunit that enables an illuminating comparison to the corresponding data from B3. We present this data in Figure 2—figure supplement 2.

*5) Discussion, end of third paragraph: Yang et al. (Yang et al., 2009) shows R995 interacting with D723, not K716, so this statement saying that the data in that publication needs revisiting is not correct.*

The reviewer is correct. We apologize for the error. The text has been altered to remove this statement.

*6) Please remove mention of 'structure' in the second paragraph of the subsection “The β1 and β3 TMD/CTD have different structures” as this refers to a structural model only.*

Throughout this paper, including the instance cited above, we have tried to be more careful in our wording to describe the molecular depictions in Figure 2 as structural models (albeit experimentally determined) rather than as a true experiment 3-D structure.

*7) The secondary structure outside the bicelle is surprising as helices do not usually form spontaneously without additional interactions; one interpretation might be that the helix is actually be binding back on to lipids. Please include some distance restraints, or residual dipolar coupling angle restraints to actually demonstrate that these secondary structure elements form in the absence of binding partners (or whether they are low propensity events as the fraying suggests). It seems unlikely that a single stranded β sheet would ever exist as is currently suggested; please include further discussion around this point.*

We have collected and now present additional data that both confirms the structural models and also illuminates them. First, we present new site-specific amide/water hydrogen exchange measurements in Figure 2 that support the continuation of the helices for both B1 and B3 in to the cytosol, but do indicate some fraying of these helices. This is supported by the R2 relaxation data mentioned earlier that is presented as Figure 2—figure supplement 2. We also now present the output of the TALOS-N chemical shift analysis that was the original basis for the secondary structure in the models as the new Figure 2—figure supplement 2. In the text we acknowledge the presence of fraying in both the B1 and B3 CT helices as well as the presence of a model degree of hinge-like motion at the TM/CT interface, which is supported by both the amide/water hydrogen exchange data and the R2 relaxation data.

We wish to make a few other points in response to this reviewer concern. The extensive paramagnetic probe data presented in this paper for the non-truncated combined TM/CT domains (Figure 5) makes it very clear that the cytosolic segments observed to be helical based on TALOS analysis of the NMR backbone chemical shifts for both B1 and B3 are *not* interacting with the membrane surface, but extend into the aqueous phase. While it is true that monomeric water soluble polypeptides usually do not form stable helices in solution, we argue that the fact that the cytoplasmic domain helices are extensions of highly stable membrane-embedded helices makes them much more stable than they would be as excised peptides in solution. That this is energetically reasonable is supported by crystal structures of many membrane proteins in which TM helices are extended well beyond the membrane. Indeed in the structure of the αIIb/β3 integrin TM/CT structure determined by the Qin lab (Armulik, Velling and Johansson, 2004) in a water/organic solvent mixture, they observed the beta3 TM helix to extend even further into the cytosol than we do even though it does not make any tertiary structure interactions with the α subunit beyond the juxtamembrane clasp. Others also have proposed structures for B3 or for the αIIb/B3 complex (based on a variety of data or modeling that include an extended B3 helix into the cytosol that is contiguous with the transmembrane domain (c.f. Zhu, Luo, Barth et al. Molecular Cell, 33, 234-249, 2009; Wegener, Partridge et al. Cell 128, 171-182 2007; Anthis, Wegener et al. EMBO J 28, 3623-3632, 2009; Metclaf, Moore et al. PNAS, 107, 22481-22486, 2010). We also emphasize that what is most interesting about results of our studies is the fact that the B1 helix is much shorter than that of B3, extending much less far into the cytosol.

*8) The Anisotropy scale on Figure 6 is either incorrect (the change in anisotropy is 0.005 which is an incredibly small amount) or is not actually measuring a change in binding. Figure 6 is the only panel that gives a reasonable change in anisotropy (0.6) even though it shows the binding is weak. This is particularly important, as this is the only data that supports a major finding of this paper (constitutive activation of a1B1 and a2B1) and therefore the findings and potential caveats requires further discussion. Please also discuss why the Bmax of the anisotropy for the KE mutant higher: if it binds less well and has not reached saturation then one would expect less anisotropy?*

*9) The fluorescence anisotropy of the α1 and α2 might be due to an issue with the α tail peptide accessing the vesicle, or entering incorrectly. Just because the binding is weak with isolated peptides they may well interact (albeit weaker) in the context of the whole integrin with the result being that it might be more easily activated, however, that is not the same as being constitutively active. The authors should include discussion of this point in the revised study.*

*10) Given the concerns over the anisotropy data being insufficient to support the notion of differential activation, we request that additional chemical shift experiments are performed using β3+αIIb and β1+α1/α2/α5 tails to permit more transparent and direct comparison across the different heterodimers and help support the key conclusions relating to activation status of each integrin.*

In response to these concerns we have completed and analyzed a complete set of NMR titrations for wild type α1, α2, and α5 binding to β1 (both wild type and K752E mutant) and wild type αIIb binding to β3 (both wild type and K716E mutant). The results are presented in Figure 6 and Figure 7 and in Table 1. As can be seen in Table 1, the NMR results are in all cases in reasonably good agreement with the fluorescence anisotropy results, particularly when different experimental temperatures are taken into account (NMR: 45 °C, fluorescence anisotropy: 35 °C). To address the reviewer’s question about why the changes in anisotropy were often very small, we note that these experiments involved titrations of on integrin subunit in bicelles with another integrin subunit in bicelles, such that the change in tumbling rate for the fluorescently labeled subunit upon heterodimer formation in bicelles was not large.

*11) The probe accessibility measurements show no change with mutation which is surprising if the lysine is buried in the membrane. This data also does not give any readout on membrane tilt; no effect is seen but it is not clear if this is due to the experiment or the mutant. Without the α tail present it is not possible to judge the overall effect of this mutation as the lysine is positioned with the potential to make a number of contacts to phenylalanines on the α tail. It is possible that the lysine makes a cation-pi bond to the delocalized electrons of the phenylalanine ring and thus in a more physiologically relevant context the mutant might be more significant. The authors should discuss this point.*

It is well established by the classical Weiner and White neutron diffraction measurements and other previous work that the concentration of water at the bilayer interface, even in the glycerol backbone region is significant. Thus, we think that the hydrogen bonding potential of either a lysine side chain or of a glutamate side chain (both of which would orient their relatively long side chains toward the aqueous phase) could be satisfied by interaction with interfacial water even though these sites are located in the membrane.

We acknowledge that our methods do not provide a readout of the tilt of the transmembrane domains, but note that this is a ubiquitous problem in the integrin field. We agree with the reviewer that cation-pi interactions between the key lysine amino group and the Phe side chains located in the adjacent α subunit could be key interactions that stabilize integrin heterodimers. However, the presence of such interactions is not established by our work and invoking this interaction cannot explain the very different functional consequences of the Lys-to-Glu mutation in β1 vs. β3 integrins. For this reason and because the manuscript is already rather long we do not take up the reviewer’s suggestion to discuss possible cation-pi interactions in the text.